# TSGM: Regular and Irregular Time-series Generation using Score-based Generative Models

## Abstract

Score-based generative models (SGMs) have demonstrated unparalleled sampling quality and diversity in numerous fields, such as image generation, voice synthesis, and tabular data synthesis, etc. Inspired by those outstanding results, we apply SGMs to synthesize time-series by learning its conditional score function. To this end, we present a conditional score network for time-series synthesis, deriving a denoising score matching loss tailored for our purposes. In particular, our presented denoising score matching loss is the first denoising score matching loss for time-series synthesis. In addition, our framework is such flexible that both regular and irregular time-series can be synthesized with minimal changes to our model design. Finally, we obtain exceptional synthesis performance on various time-series datasets, achieving state-of-the-art sampling diversity and quality.

## 1 Introduction

Time-series frequently occurs in our daily life, e.g., stock data, climate data, and so on. Especially, time-series forecasting and classification are popular research topics in the field of machine learning Ahmed et al. (2010); Fu (2011); Ismail Fawaz et al. (2019). In many cases, however, time-series samples are incomplete and/or the number of samples is insufficient, in which case training machine learning models cannot be fulfilled in a robust way. To overcome the limitation, time-series synthesis has been studied actively recently (Chen et al., 2018; Dash et al., 2020). These synthesis models have been designed in various ways, including variational autoencoders (VAEs) and generative adversarial networks (GANs) (Desai et al., 2021; Yoon et al., 2019; Jeon et al., 2022). In particular, real-world time series is often irregular, i.e., the inter-arrival time between observations is not fixed and/or some observations can be missing. In addition, datasets like Physionet (Silva et al., 2012) deliberately obscure certain features to protect patient privacy, posing challenges for training and analyses. In such a case, releasing synthesized time series is needed for alleviating privacy concerns, but it is challenging to solve (Jeon et al., 2022).

Despite the previous efforts to generate time-series using GANs and VAEs, according to our survey, there is no research using SGMs for this purpose. Therefore, we extend SGMs into the field of time-series synthesis[1]. Unlike image generation, where each image can be generated independently, in time-series generation, each time-series observation is generated in consideration of its previously generated observations. To this end, we propose the method of **T**ime-series generation using conditional **S**core-based **G**enerative **M**odel (TSGM), which consists of three neural networks, i.e., an encoder, a score network, and a decoder (see Figure 2).

**Score-based time-series synthesis** Our proposed method can be characterized by the following two parts. First, we design a conditional score network on time-series, which learns the gradient of the conditional log-likelihood with respect to the sequential order of time-series. Second, we also design a denoising score matching loss for our conditional time-series generation and prove its correctness.

---

[1]There exist time-series diffusion models for forecasting and imputation (Rasul et al., 2021; Tashiro et al., 2021). However, our time-series synthesis is technically different from i) time-series forecasting, which forecasts future observations given past observations, and ii) time-series imputation, which given a time-series sample with missing elements infers those missing ones. We discuss the differences in Appendix B and C.

Table 1: The table illustrates how many medals each method gets across all datasets and evaluation metrics, based on the generation evaluation scores presented in Table 3, Table 4, and Table 16. Our method with the two specific types, TSGM-VP and TSGM-subVP, achieves superior generation performance compared to baselines.

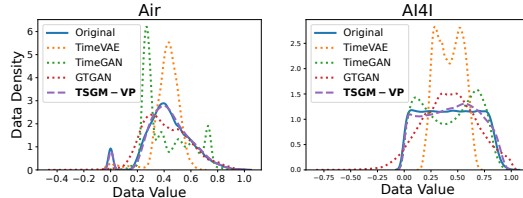

Figure 1: The KDE plots show the estimated distributions of original data and ones generated by several methods in the Air and AI4I datasets — we ignore time stamps for drawing these distributions. Unlike baseline methods, the distribution of TSGM-VP is almost identical to the original one. These figures provide an evidence of the excellent generation quality and diversity of our method. For TSGM-subVP, similar results are observed. Refer to Appendix L for additional visualizations

|  | Olympic Rankings | | | | | |
| Method | Gold | | Silver | | Bronze | |
|  | Regular | Irregular | R | I | R | I |
| TSGM-VP | 4 | 11 | 4 | 11 | 0 | 1 |
| TSGM-subVP | 6 | 16 | 1 | 7 | 1 | 0 |
| TimeGAN | 1 | 0 | 0 | 0 | 1 | 0 |
| TimeVAE | 0 | 0 | 0 | 0 | 1 | 4 |
| GT-GAN | 0 | 1 | 1 | 1 | 2 | 16 |

**Regular vs. irregular time-series synthesis** In addition, our method is such flexible that both regular and irregular time-series samples can be synthesized with minimal changes to our model design. For synthesizing regular time series, we use a recurrent neural network-based encoder and decoder. Continuous-time methods, such as neural controlled differential equations (Kidger et al., 2020) and GRU-ODE (Brouwer et al., 2019), can be used as our encoder and decoder for synthesizing irregular time series (see Section 3.2 and Appendix I).

We conduct in-depth experiments with 4 real-world datasets under regular and irregular settings — for the irregular settings, we randomly drop 30%, 50%, and 70% of observations from regular time-series, which means our problem statement is only with respect to missing data in an otherwise regularly sampled dataset. Therefore, we test with in total 16 different settings, i.e., 4 datasets for one regular and three irregular settings. Our specific choices of 8 baselines include almost all existing types of time-series generative paradigms, ranging from VAEs to GANs. In Table 1 and Figure 1, we compare our method to the baselines, ranking methods by their evaluation scores and estimating data distribution by kernel density estimation (KDE). We also visualize real and generated time-series samples onto a latent space using t-SNE (van der Maaten & Hinton, 2008) in Figure 3. Our proposed method shows the best generation quality in almost all cases. Furthermore, the t-SNE and KDE visualization results provide intuitive evidence that our method's generation diversity is also superior to that of the baselines. Our contributions are summarized as follows:

1. We, for the first time, propose an SGM-based time-series synthesis method. Although there exist diffusion-based time-series forecasting and imputation methods, our target score function and its denoising score matching loss definition are totally different from other baselines. We highlight the difference and inappropriateness of the forecasting and imputation methods to our task in Section B.

2. We, therefore, derive our own denoising score matching loss considering the fully recurrent nature of our time-series generation, i.e., recursively generate complete time-series observations from scratch.

3. We conduct comprehensive experiments with 4 real-world datasets and 8 baselines under one regular and three irregular settings since our method supports both regular and irregular time-series. Overall, our proposed method shows the best generation quality and diversity.

## 2 RELATED WORK AND PRELIMINARIES

### 2.1 SCORE-BASED GENERATIVE MODELS

SGMs offer several advantages over other generative models, including their higher generation quality and diversity. SGMs follow a two-step process, wherein i) gaussian noises are continuously added to a sample and ii) then removed to recover a new sample. These processes are known as the forward

and reverse processes, respectively. In this section, we provide a brief overview of the original SGMs in Song et al. (2021), which will be adapted for the time-series generation tasks.

### 2.1.1 FORWARD AND REVERSE PROCESS

At first, SGMs add noises with the following stochastic differential equation (SDE):

$$d\mathbf{x}^s = \mathbf{f}(s, \mathbf{x}^s)ds + g(s)d\mathbf{w}, \quad s \in [0, 1], \tag{1}$$

where $\mathbf{w} \in \mathbb{R}^{\dim(\mathbf{x})}$ is a multi-dimensional Brownian motion, $\mathbf{f}(s, \cdot) : \mathbb{R}^{\dim(\mathbf{x})} \to \mathbb{R}^{\dim(\mathbf{x})}$ is a vector-valued drift term, and $g : [0, 1] \to \mathbb{R}$ is a scalar-valued

Table 2: Comparison of drift and diffusion terms. $\sigma(s)$ means positive noise values which are increasing, and $\beta(s)$ denotes noise values in [0,1], which are used in Song & Ermon (2019); Ho et al. (2020)

| SDE | drift ($\mathbf{f}$) | diffusion ($g$) |
|---|---|---|
| VE | $0$ | $\sqrt{\frac{d\sigma^2(s)}{ds}}$ |
| VP | $-\frac{1}{2}\beta(s)\mathbf{x}^s$ | $\sqrt{\beta(s)}$ |
| subVP | $-\frac{1}{2}\beta(s)\mathbf{x}^s$ | $\sqrt{\beta(s)(1 - e^{-2\int_0^s \beta(t)dt})}$ |

diffusion function. Here after, we define $\mathbf{x}^s$ as a noisy sample diffused at time $s \in [0, 1]$ from an original sample $\mathbf{x} \in \mathbb{R}^{\dim(\mathbf{x})}$. Therefore, $\mathbf{x}^s$ can be understood as a stochastic process following the SDE. There are several options for $\mathbf{f}$ and $g$: variance exploding(VE), variance preserving(VP), and subVP. Song et al. (2021) proved that VE and VP are continuous generalizations of the two discrete diffusion methods: one in Song & Ermon (2019) and the other in Sohl-Dickstein et al. (2015); Ho et al. (2020). The subVP method shows, in general, better negative log-likelihood (NLL) according to Song et al. (2021). We describe the exact form of each SDE in Table 2 with detailed explanation in Appendix N. Note that we only use the VP and subVP-based TSGM in our experiments and exclude the VE-based one for its inferiority for time series synthesis in our experiments.

SGMs run the forward SDE with a sufficiently large number of steps to make sure that the diffused sample converges to a Gaussian distribution at the final step. The score network $M_\theta(s, \mathbf{x}^s)$ learns the gradient of the log-likelihood $\nabla_{\mathbf{x}^s} \log p(\mathbf{x}^s)$, which will be used in the reverse process.

For the forward SDE, there exists the following corresponding reverse SDE (Anderson, 1982):

$$d\mathbf{x}^s = [\mathbf{f}(s, \mathbf{x}^s) - g^2(s)\nabla_{\mathbf{x}^s} \log p(\mathbf{x}^s)]ds + g(s)d\bar{\mathbf{w}}. \tag{2}$$

The formula suggests that if knowing the score function, $\nabla_{\mathbf{x}^s} \log p(\mathbf{x}^s)$, we can recover real samples from the prior distribution $p_1(\mathbf{x}) \sim \mathcal{N}(\mu, \sigma^2)$, where $\mu, \sigma$ vary depending on the forward SDE type.

### 2.1.2 TRAINING PROCESS

In order for the model $M$ to learn the score function, the model has to optimize the following loss function:

$$L(\theta) = \mathbb{E}_s\{\lambda(s)\mathbb{E}_{\mathbf{x}^s}[\|M_\theta(s, \mathbf{x}^s) - \nabla_{\mathbf{x}^s} \log p(\mathbf{x}^s)\|_2^2]\}, \tag{3}$$

where $s$ is uniformly sampled over $[0, 1]$ with an appropriate weight function $\lambda(s) : [0, 1] \to \mathbb{R}$. However, using the above formula is computationally prohibitive (Hyvärinen, 2005; Song et al., 2019). Thanks to Vincent (2011), the loss can be substituted with the following denoising score matching loss:

$$L^*(\theta) = \mathbb{E}_s\{\lambda(s)\mathbb{E}_{\mathbf{x}^0}\mathbb{E}_{\mathbf{x}^s|\mathbf{x}^0}[\|M_\theta(s, \mathbf{x}^s) - \nabla_{\mathbf{x}^s} \log p(\mathbf{x}^s|\mathbf{x}^0)\|_2^2]\}. \tag{4}$$

Since SGMs use an affine drift term, the transition kernel $p(\mathbf{x}^s|\mathbf{x}^0)$ follows a certain Gaussian distribution (Särkkä & Solin, 2019) and therefore, $\nabla_{\mathbf{x}^s} \log p(\mathbf{x}^s|\mathbf{x}^0)$ can be analytically calculated.

### 2.2 TIME-SERIES GENERATION

Let $\mathbf{x}_{1:N}$ be a time-series sample which consists of $N$ observations. In order to synthesize time-series $\mathbf{x}_{1:N}$, unlike other generation tasks, we must generate each observation $\mathbf{x}_n$ at sequential order $n \in \{2, ..., N\}$ considering its previous history $\mathbf{x}_{1:n-1}$. One can train neural networks to learn the conditional likelihood $p(\mathbf{x}_n|\mathbf{x}_{1:n-1})$ and generate each $\mathbf{x}_n$ recursively using it. There are several time-series generation papers, and we introduce their ideas.

TimeVAE (Desai et al., 2021) is a variational autoencoder to synthesize time-series data. This model can provide interpretable results by reflecting temporal structures such as trend and seasonality in

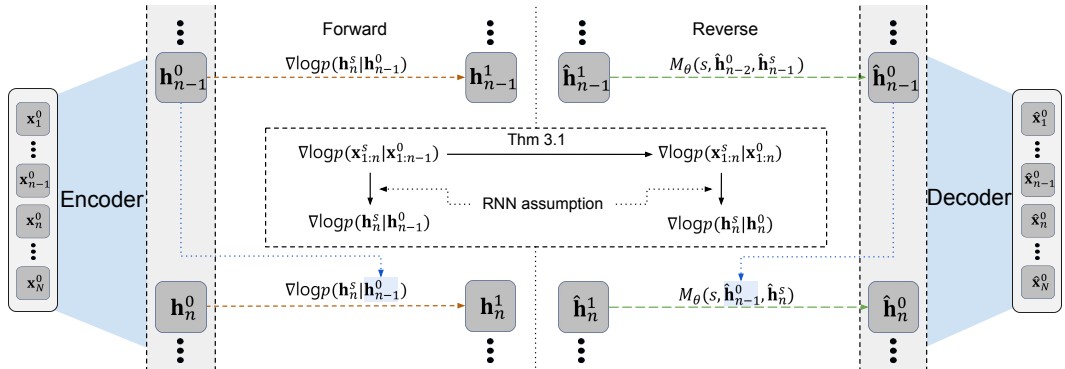

Figure 2: The overall workflow of TSGM (see Section 3.3). Our original learning objective is to approximate $\nabla \log p(\mathbf{x}_{1:n}^s | \mathbf{x}_{1:n-1}^0)$, which is computationally prohibitive, with the conditional score network $M_\theta(s, \mathbf{x}_{1:n}^s, \mathbf{x}_{1:n-1}^0)$ using an MSE loss. We then prove in Thm. 3.1 that learning $\nabla \log p(\mathbf{x}_{1:n}^s | \mathbf{x}_{1:n}^0)$ is equivalent to $\nabla \log p(\mathbf{x}_{1:n}^s | \mathbf{x}_{1:n-1}^0)$ for $\theta$ of $M_\theta$ in the MSE loss, i.e., their optimal model parameter $\theta$ is identical. At the end, our score network $M_\theta(s, \mathbf{h}_n^s, \mathbf{h}_{n-1}^0)$ learns $\nabla \log p(\mathbf{h}_n^s | \mathbf{h}_n)$ since RNNs can encode $\mathbf{x}_{1:n}^0$ and $\mathbf{x}_{1:n-1}^0$ into their hidden states $\mathbf{h}_n^0$ and $\mathbf{h}_{n-1}^0$, respectively.

the generation process. CTFP (Deng et al., 2020) is a well-known normalizing flow model. It can treat both regular and irregular time-series data by a deformation of the standard Wiener process. TimeGAN (Yoon et al., 2019) uses a GAN architecture to generate time-series. First, it trains an encoder and decoder, which transform a time-series sample $\mathbf{x}_{1:N}$ into latent vectors $\mathbf{h}_{1:N}$ and recover them by using a recurrent neural network (RNN). Next, it trains a generator and discriminator pair on latent space, by minimizing the discrepancy between an estimated and true distribution, i.e., $\hat{p}(\mathbf{x}_n | \mathbf{x}_{1:n-1})$ and $p(\mathbf{x}_n | \mathbf{x}_{1:n-1})$. Since it uses an RNN-based encoder, it can efficiently learn the conditional likelihood $p(\mathbf{x}_n | \mathbf{x}_{1:n-1})$ by treating it as $p(\mathbf{h}_n | \mathbf{h}_{n-1})$, since $\mathbf{h}_n \sim \mathbf{x}_{1:n}$ under the regime of RNNs. Therefore, it can generate each observation $\mathbf{x}_n$ considering its previous history $\mathbf{x}_{1:n-1}$. However, GAN-based generative models are vulnerable to the issue of mode collapse (Xiao et al., 2022) and unstable behavior problems during training (Chu et al., 2020). GT-GAN (Jeon et al., 2022) attempted to solve the problems by incorporating an invertible neural network-based generator into its framework. There also exist GAN-based methods to generate other types of sequential data, e.g., video, sound, etc (Esteban et al., 2017; Mogren, 2016; Xu et al., 2020; Donahue et al., 2019). In our experiments, we also use them as our baselines for thorough evaluations.

## 3 PROPOSED METHOD

Our proposed TSGM consists of three networks: an encoder, a decoder, and a conditional score network (cf. Fig. 2). Firstly, we train the encoder and the decoder to connect between time-series samples and a latent space. Next, using the pre-trained encoder and decoder, we train the conditional score network on the latent space. The conditional score network will be used for sampling fake time-series on the latent space.

### 3.1 PROBLEM FORMULATION

Let $\mathcal{X}$ and $\mathcal{H}$ denote a data space and a latent space, respectively. We define $\mathbf{x}_{1:N}$ as a time-series sample with a sequential length of $N$, and $\mathbf{x}_n$ is a multi-dimensional observation of $\mathbf{x}_{1:N}$ at sequential order $n$. Similarly, $\mathbf{h}_{1:N}$ (resp. $\mathbf{h}_n$) denotes an embedded time series (resp. an embedded observation).

Each observation $\mathbf{x}_n$ can be represented as a pair of time and features, i.e., $\mathbf{x}_n = (t_n, \mathbf{u}(t_n))$, where $t_n \in \mathbb{R}_{\geq 0}$ is a time stamp of feature $\mathbf{u}(t_n) \in \mathbb{R}^{\dim(\mathbf{u})}$, and $\dim(\mathbf{u})$ is a feature dimension. $\mathcal{X}$ can be classified into two types: regular time-series and irregular time-series. For creating irregularly sampled time-series, we randomly drop 30%, 50%, and 70% of observations from regular time-series, which means our problem statement is only with respect to missing data in an otherwise regularly sampled dataset.

## 3.2 ENCODER AND DECODER

The encoder and decoder have the task of mapping time-series data to a latent space and vice versa. We define $e$ and $d$ as an encoding function mapping $\mathcal{X}$ to $\mathcal{H}$ and a decoding function mapping $\mathcal{H}$ to $\mathcal{X}$, respectively. In regular time-series generation, we assume RNN-based encoder and decoder. It is hyperparameter to choose which RNN would be used to the encoder $e$ and the decoder $d$, and we utilize gated recurrent units (GRUs) as Yoon et al. (2019) did. Since we use RNNs, both $e$ and $d$ are defined recursively as follows:

$$\mathbf{h}_n = e(\mathbf{h}_{n-1}, \mathbf{x}_n), \qquad \hat{\mathbf{x}}_n = d(\mathbf{h}_n), \tag{5}$$

where $\hat{\mathbf{x}}_n$ denotes a reconstructed time-series sample at sequential order $n$. In irregular time-series generation, we suppose a Neural CDE-based encoder and a GRU-ODE-based decoder, which are famous for dealing with irregular time-series (Kidger et al., 2020; Brouwer et al., 2019). We describe details of how continuous-time methods can be used in Appendix I.

After embedding real time-series data onto a latent space, we can train the conditional score network with its conditional log-likelihood, whose architecture is described in Appendix J.2. The encoder and decoder are pre-trained before our main training.

## 3.3 TRAINING OBJECTIVE FUNCTION

To generate time-series, our score network has to learn the conditional score function as we mentioned in Section 2.2. More precisely, given past observations $\mathbf{x}_{1:n-1}$, our goal is to learn the conditional score function, $\nabla_{\mathbf{x}_{1:n}^s} \log p(\mathbf{x}_{1:n}^s | \mathbf{x}_{1:n-1}^0)$ where $s \in [0, 1]$ is a diffusion step. However, considering the total sequence $\mathbf{x}_{1:n}$ is computationally expensive, so we train an autoencoder to replace it with its latent feature, i.e., $\mathbf{h}_n \sim \mathbf{x}_{1:n}$, as previous works did (cf. Section 2.2). Therefore, our loss function is composed of two parts: one for the autoencoder and the other for the score network.

**Loss for autoencoder** We use two training objective functions. First, we train the encoder and the decoder using $L_{ed}$. Let $\mathbf{x}_{1:N}^0$ and $\hat{\mathbf{x}}_{1:N}^0$ denote an real time-series sample and its reconstructed copy by the encoder-decoder process, respectively. Each $\mathbf{x}_{1:N}^0$ are selected from a probability distribution $p(\mathbf{x}_{1:N}^0)$. Then, $L_{ed}$ denotes the following MSE loss between $\mathbf{x}_{1:N}^0$ and its reconstructed copy $\hat{\mathbf{x}}_{1:N}^0$:

$$L_{ed} = \mathbb{E}_{\mathbf{x}_{1:N}^0}[\|\hat{\mathbf{x}}_{1:N}^0 - \mathbf{x}_{1:N}^0\|_2^2]. \tag{6}$$

**Loss for score network** Next, we define another loss $L_{score}^{\mathcal{H}}$ in equation 12 to train the conditional score network $M_\theta$, which is one of our main contributions. In order to derive the training loss $L_{score}^{\mathcal{H}}$ from the initial loss definition $L_1$, we describe its step-by-step derivation procedure. At sequential order $n$ in $\{1, ..., N\}$, we diffuse $\mathbf{x}_{1:n}^0$ through a sufficiently large number of steps of the forward SDE (1) to a Gaussian distribution. Let $\mathbf{x}_{1:n}^s$ denotes a diffused sample at step $s \in [0, 1]$ from $\mathbf{x}_{1:n}^0$. Then the conditional score network $M_\theta(s, \mathbf{x}_{1:n}^s, \mathbf{x}_{1:n-1}^0)$ can be trained to learn the gradient of the conditional log-likelihood with the following $L_1$ loss:

$$L_1 = \mathbb{E}_s \mathbb{E}_{\mathbf{x}_{1:N}^0} \left[ \sum_{n=1}^{N} \lambda(s) l_1(n, s) \right], \tag{7}$$

$$\text{where} \quad l_1(n, s) = \mathbb{E}_{\mathbf{x}_{1:n}^s} \left[ \left\| M_\theta(s, \mathbf{x}_{1:n}^s, \mathbf{x}_{1:n-1}^0) - \nabla_{\mathbf{x}_{1:n}^s} \log p(\mathbf{x}_{1:n}^s | \mathbf{x}_{1:n-1}^0) \right\|_2^2 \right]. \tag{8}$$

In the above definition, $\nabla_{\mathbf{x}_{1:n}^s} \log p(\mathbf{x}_{1:n}^s | \mathbf{x}_{1:n-1}^0)$, where $\mathbf{x}_i^0$ depends on $\mathbf{x}_{1:i-1}^0$ for each $i \in \{2, ..., n\}$, is designed specially for time-series generation. Note that for our training, $\mathbf{x}_{1:n}^s$ is sampled from $p(\mathbf{x}_{1:n}^s | \mathbf{x}_{1:n-1}^0)$, and $s$ is uniformly sampled from $[0, 1]$.

However, using the above formula, which is a naïve score matching on time-series, is computationally prohibitive (Hyvärinen, 2005; Song et al., 2019). Thanks to the following theorem, the more efficient denoising score loss $L_2$ can be defined.

**Theorem 3.1** (Denoising score matching on time-series). $l_1(n, s)$ *can be replaced with the following* $l_2(n, s)$

$$l_2(n, s) = \mathbb{E}_{\mathbf{x}_n^0} \mathbb{E}_{\mathbf{x}_{1:n}^s} \left[ \left\| M_\theta(s, \mathbf{x}_{1:n}^s, \mathbf{x}_{1:n-1}^0) - \nabla_{\mathbf{x}_{1:n}^s} \log p(\mathbf{x}_{1:n}^s | \mathbf{x}_{1:n}^0) \right\|_2^2 \right], \tag{9}$$

*where i) $\mathbf{x}_n^0$ and $\mathbf{x}_{1:n}^s$ are sampled from $p(\mathbf{X}_n^0|\mathbf{X}_{1:n-1}^0)$ and $p(\mathbf{X}_{1:n}^s|\mathbf{X}_{1:n}^0)$ to calculate the nested expectations; ii) $\nabla_{\mathbf{x}_{1:n}^s} \log p(\mathbf{X}_{1:n}^s|\mathbf{X}_{1:n-1}^0)$ of $L_1$ is changed to $\nabla_{\mathbf{x}_{1:n}^s} \log p(\mathbf{X}_{1:n}^s|\mathbf{X}_{1:n}^0)$. Therefore, we can use an alternative objective, $L_2 = \mathbb{E}_s \mathbb{E}_{\mathbf{X}_{1:N}^0} \left[ \sum_{n=1}^N \lambda(s) l_2(n, s) \right]$ instead of $L_1$.* □

However, $L_2$ still has a problem since it has to sample each $\mathbf{x}_n^0$ using $p(\mathbf{x}_n^0|\mathbf{x}_{1:n-1}^0)$ every time and therefore, we describe another corollary and thereby propose $L_{score}$.

**Corollary 3.2.** *Our target objective function, $L_{score}$, is defined as follows:*

$$L_{score} = \mathbb{E}_s \mathbb{E}_{\mathbf{X}_{1:N}^0} \left[ \sum_{n=1}^N \lambda(s) l_2^\star(n, s) \right], \tag{10}$$

*where $l_2^\star(n, s) = \mathbb{E}_{\mathbf{X}_{1:n}^s} \left[ \left\| M_\theta(s, \mathbf{X}_{1:n}^s, \mathbf{X}_{1:n-1}^0) - \nabla_{\mathbf{x}_{1:n}^s} \log p(\mathbf{X}_{1:n}^s|\mathbf{X}_{1:n}^0) \right\|_2^2 \right].$ \tag{11}*

*Then, $L_2 = L_{score}$ is satisfied.* □

Note that the only difference between $L_{score}$ and $L_2$ is the existence of expectation with respect to $\mathbf{x}_n^0$. As such, $L_{score}$ provides more amenable training procedures than $L_2$ since it doesn't need to additionally sample each $\mathbf{x}_n^0$. Moreover, they are equivalent according to the corollary.

Our pre-trained the encoder and decoder encode data autoregressively as Equation (5) shows, and it is same in the irregular time-series case, too. So the encoder can embed $\mathbf{x}_{1:n}^0$ into $\mathbf{h}_n^0 \in \mathcal{H}$. Ideally, $\mathbf{h}_n^0$ involves the entire information of $\mathbf{x}_{1:n}^0$. Therefore, $L_{score}$ can be re-written as follows with the embeddings in the latent space:

$$L_{score}^{\mathcal{H}} = \mathbb{E}_s \mathbb{E}_{\mathbf{h}_{1:N}^0} \sum_{n=1}^N \left[ \lambda(s) l_3(n, s) \right], \tag{12}$$

with $l_3(n, s) = \mathbb{E}_{\mathbf{h}_n^s} \left[ \left\| M_\theta(s, \mathbf{h}_n^s, \mathbf{h}_{n-1}^0) - \nabla_{\mathbf{h}_n^s} \log p(\mathbf{h}_n^s|\mathbf{h}_n^0) \right\|_2^2 \right]$. $L_{score}^{\mathcal{H}}$ is what we use for our experiments (instead of $L_{score}$). Until now, we introduced our target objective functions, $L_{ed}$ and $L_{score}^{\mathcal{H}}$. We note that we use exactly the same weight $\lambda(s)$ as that in Song et al. (2021). Related proofs are given in Appendix A.

### 3.4 Training and Sampling Procedures

**Training method** We explain details of our training method. At first, we pre-train both the encoder and decoder using $L_{ed}$. After pre-training them, we train the conditional score network. When training the latter one, we use the embedded hidden vectors produced by the encoder. After encoding an input $\mathbf{x}_{1:N}^0$, we obtain its latent vectors $\mathbf{h}_{1:N}^0$ — we note that each hidden vector $\mathbf{h}_n^0$ has all the previous information from 1 to $n$ for the RNN-based encoder's autoregressive property as shown in the Equation (5). We use the following forward process (Song et al., 2021), where $n$ means the sequence order of the input time-series, and $s$ denotes the time (or step) of the diffusion step :

$$d\mathbf{h}_n^s = \mathbf{f}(s, \mathbf{h}_n^s)ds + g(s)d\mathbf{w}, \qquad s \in [0, 1]. \tag{13}$$

Note that we only use the VP and subVP-based TSGM in our experiments and exclude the VE-based one for its inferiority for time series synthesis in our experiments. During the forward process, the conditional score network reads the pair $(s, \mathbf{h}_n^s, \mathbf{h}_{n-1}^0)$ as input and thereby, it can learn the conditional score function $\nabla \log p(\mathbf{h}_n^s|\mathbf{h}_{n-1}^0)$ by using $L_{score}^{\mathcal{H}}$, where $\mathbf{h}_0^0 = \mathbf{0}$.

**Sampling method** After the training procedure, we use the following conditional reverse process:

$$d\mathbf{h}_n^s = [\mathbf{f}(s, \mathbf{h}_n^s) - g^2(s)\nabla_{\mathbf{h}_n^s} \log p(\mathbf{h}_n^s|\mathbf{h}_{n-1}^0)]ds + g(s)d\bar{\mathbf{w}}, \tag{14}$$

where $s$ is uniformly distributed over $[0, 1]$, theoretically. Although we assume the noises are added continuously by following the forward SDE (1), we set 1000 steps for denoising procedure on sampling, which is a default value same with Song et al. (2021), meaning $s \in \{0, 1 \cdot 10^{-3}, \ldots, 1\}$. Therefore, we uniformly choose $s$ over $[0, 1]$ on training and recover data by the above discrete denoising steps. The conditional score function in this process can be replaced with the trained score network $M_\theta(s, \mathbf{h}_n^s, \mathbf{h}_{n-1}^0)$. The detailed sampling method is as follows:

Table 3: Experimental results for the regular time-series with respect to the discriminative and predictive scores. The best scores are in boldface.

| Method | Disc. | | | | Pred. | | | |
|---|---|---|---|---|---|---|---|---|
| | Stocks | Energy | Air | AI4I | Stocks | Energy | Air | AI4I |
| TSGM-VP | .022±.005 | .221±.025 | **.122±.014** | .147±.005 | .037±.000 | .257±.000 | **.005±.000** | **.217±.000** |
| TSGM-subVP | **.021±.008** | **.198±.025** | .127±.010 | .150±.010 | **.037±.000** | **.252±.000** | .005±.000 | .217±.000 |
| T-Forcing | .226±.035 | .483±.004 | .404±.020 | .435±.025 | .038±.001 | .315±.005 | .008±.000 | .242±.001 |
| P-Forcing | .257±.026 | .412±.006 | .484±.007 | .443±.026 | .043±.001 | .303±.006 | .021±.000 | .220±.000 |
| TimeGAN | .102±.031 | .236±.012 | .447±.017 | **.070±.009** | .038±.001 | .273±.004 | .017±.004 | .253±.002 |
| RCGAN | .196±.027 | .336±.017 | .459±.104 | .234±.015 | .040±.001 | .292±.005 | .043±.000 | .224±.001 |
| C-RNN-GAN | .399±.028 | .499±.001 | .499±.000 | .499±.001 | .038±.000 | .483±.005 | .111±.000 | .340±.006 |
| TimeVAE | .175±.031 | .498±.006 | .381±.037 | .446±.024 | .042±.002 | .268±.004 | .013±.002 | .233±.010 |
| COT-GAN | .285±.030 | .498±.000 | .423±.001 | .411±.018 | .044±.000 | .260±.000 | .024±.001 | .220±.000 |
| CTFP | .499±.000 | .500±.000 | .499±.000 | .499±.001 | .084±.005 | .469±.008 | .476±.235 | .412±.024 |
| GT-GAN | .077±.031 | .221±.068 | .413±.001 | .394±.090 | .040±.000 | .312±.002 | .007±.000 | .239±.000 |
| Original | N/A | N/A | N/A | N/A | .036±.001 | .250±.003 | .004±.000 | .217±.000 |

1. At first, we sample $\mathbf{z}_1$ from a Gaussian prior distribution and set $\mathbf{h}_1^1 = \mathbf{z}_1$ and $\mathbf{h}_0^0 = \mathbf{0}$. We then generates an initial observation $\hat{\mathbf{h}}_1^0$ by denoising $\mathbf{h}_1^1$ following the conditional reverse process with $M_\theta(s, \mathbf{h}_n^s, \mathbf{h}_0^0)$ via the *predictor-corrector* method (Song et al., 2021).

2. We repeat the following computation for every $2 \leq n \leq N$, i.e., recursive generation. We sample $\mathbf{z}_n$ from a Gaussian prior distribution and set $\mathbf{h}_n^1 = \mathbf{z}_n$ for $n \in \{2, ..., N\}$. After reading the previously generated samples $\hat{\mathbf{h}}_{n-1}^0$, we then denoise $\mathbf{h}_n^1$ following the conditional reverse process with $M_\theta(s, \mathbf{h}_n^s, \hat{\mathbf{h}}_{n-1}^0)$ to generate $\hat{\mathbf{h}}_n^0$ via the *predictor-corrector* method.

Once the sampling procedure is finished, we can reconstruct $\hat{\mathbf{x}}_{1:N}^0$ from $\hat{\mathbf{h}}_{1:N}^0$ using the trained decoder at once.

## 4 EXPERIMENTS

### 4.1 EXPERIMENTAL ENVIRONMENTS

#### 4.1.1 BASELINES AND DATASETS

In the case of the regular time-series generation, we use 4 real-world datasets from various fields with 8 baselines. For the irregular time-series generation, we randomly remove some observations from each time-series sample with 30%, 50%, and 70% missing rates, which means our problem statement is only with respect to missing data in an otherwise regularly sampled dataset. Therefore, we totally treat 16 datasets, i.e., 4 datasets with one regular and three irregular settings, and 8 baselines.

Our collection of baselines covers almost all existing types of time-series synthesis methods, ranging from autoregressive generative models to normalizing flows, VAEs and GANs. For the baselines, we reuse their released source codes in their official repositories and rely on their designed training and model selection procedures. If a baseline does not support irregular time-series synthesis, we replace its RNN encoder with GRU-D (Che et al., 2016) modified from GRUs to deal with irregular time-series (see Appendix O for detailed explanation). For those that do not use an RNN-based encoder, we add GRU-D in front of the encoder, such as TimeVAE and COT-GAN. Therefore, all baselines are tested for the regular and irregular environments. We refer to Appendix E for the detailed descriptions on our datasets, baselines, and Appendix G for other software/hardware environments.

#### 4.1.2 EVALUATION METRICS

In the image generation domain, researchers have evaluated the *fidelity* and the *diversity* of models by using the Fréchet inception distance (FID) and inception score (IS). On the other hand, to measure the fidelity and the diversity of synthesized time-series samples, we use the following predictive score and the discriminative score as in (Yoon et al., 2019; Jeon et al., 2022). We strictly follow the evaluation protocol agreed by the time-series research community (Yoon et al., 2019; Jeon et al.,

Table 4: Experimental results for the irregular time-series with 30% missing rate. Results for higher missing rates in Table 16 of Appendix K.

| Method | Disc. | | | | Pred. | | | |
|---|---|---|---|---|---|---|---|---|
| | Stocks | Energy | Air | AI4I | Stocks | Energy | Air | AI4I |
| TSGM-VP | .062±.018 | **.294±.007** | **.190±.042** | .142±.048 | **.012±.002** | **.049±.001** | **.042±.002** | .067±.013 |
| TSGM-subVP | **.025±.009** | .326±.008 | .240±.018 | **.121±.082** | .012±.001 | .049±.001 | .044±.004 | **.061±.001** |
| T-Forcing-D | .409±.051 | .347±.046 | .458±.122 | .493±.018 | .027±.002 | .090±.001 | .112±.004 | .147±.010 |
| P-Forcing-D | .480±.060 | .491±.020 | .494±.012 | .430±.061 | .079±.008 | .147±.001 | .101±.003 | .134±.005 |
| TimeGAN-D | .411±.040 | .479±.010 | .500±.001 | .500±.000 | .105±.053 | .248±.024 | .325±.005 | .251±.010 |
| RCGAN-D | .500±.000 | .500±.000 | .500±.000 | .500±.000 | .523±.020 | .409±.020 | .342±.018 | .329±.037 |
| C-RNN-GAN-D | .500±.000 | .500±.000 | .500±.000 | .450±.150 | .345±.002 | .440±.000 | .354±.060 | .400±.026 |
| TimeVAE-D | .423±.088 | .382±.124 | .373±.191 | .384±.086 | .207±.014 | .139±.004 | .105±.002 | .144±.003 |
| COT-GAN-D | .499±.001 | .500±.000 | .500±.000 | .500±.000 | .274±.000 | .427±.000 | .451±.000 | .570±.000 |
| CTFP | .500±.000 | .500±.000 | .500±.000 | .499±.001 | .070±.009 | .499±.000 | .060±.027 | .424±.002 |
| GT-GAN | .251±.097 | .333±.063 | .454±.029 | .435±.018 | .077±.031 | .221±.068 | .064±.002 | .087±.013 |
| Original | N/A | N/A | N/A | N/A | .011±.002 | .045±.001 | .044±.006 | .059±.001 |

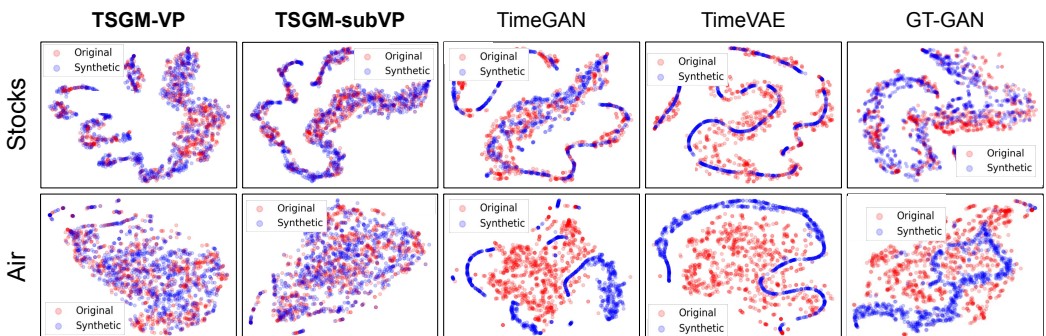

Figure 3: t-SNE plots for TSGM (1st and 2nd columns), TimeGAN (3rd columns), TimeVAE (4th columns), GT-GAN (5th columns) in Stocks and Air datasets. Red and blue dots mean original and synthesized samples, respectively. Refer to Appendix L for addition visualizations

2022). Both metrics are designed in a way that lower values are preferred. We run each generative method 10 times with different seeds, and report its mean and standard deviation of the following discriminative and predictive scores:

i) *Predictive Score*: We use the predictive score to evaluate whether a generative model can successfully reproduce the temporal properties of the original data. To do this, we first train a popular LSTM-based sequence model for time-series forecasting with synthesized samples. The performance of this predictive model is measured as the mean absolute error (MAE) on the original test data. This kind of evaluation paradigm is called as train-synthesized-test-real (TSTR) in the literature.

ii) *Discriminative Score*: In order to assess how similar the original and generated samples are, we train a 2-layer LSTM model that classifies the real/fake samples into two classes, real or fake. We use the performance of the trained classifier on the test data as the discriminative score. Therefore, lower discriminator scores mean real and fake samples are similar.

## 4.2 EXPERIMENTAL RESULTS

At first, on the regular time-series generation, Table 3 shows that our method achieves remarkable results, outperforming TimeGAN and GT-GAN except only for the discriminative score on AI4I. Especially, for Stock, Energy, and Air, TSGM exhibits overwhelming performance by large margins for the discriminative score. Moreover, for the predictive score, TSGM performs the best and obtains almost the same scores as that of the original data, which indicates that generated samples from TSGM preserve all the predictive characteristics of the original data.

Next, on the irregular time-series generation, we give the result with the 30% missing rate setting on Table 4 and other results in Appendix K. TSGM also defeats almost all baselines by large margins on both the discriminative and predictive scores. Interestingly, VP generates poorer data as the missing rate grows up, while subVP synthesizes better one.

Table 5: Sensitivity results on the depth of $M_\theta$ and the number of sampling steps. Our default TSGM has a depth of 4 and its number of sampling steps is 1,000. For other omitted datasets, we observe similar patterns.

| Method | TSGM | | Depth of 3 | | 500 steps | | 250 steps | | 100 steps | |
|---|---|---|---|---|---|---|---|---|---|---|
| SDE | VP | subVP | VP | subVP | VP | subVP | VP | subVP | VP | subVP |
| Disc. Stocks | .022±.005 | .021±.008 | .022±.004 | **.020±.007** | .025±.005 | .020±.004 | .067±.009 | .022±.009 | .202±.013 | .023±.005 |
| Disc. Energy | .221±.025 | .198±.025 | **.175±.009** | .182±.009 | .259±.003 | .248±.002 | .250±.003 | .247±.002 | .325±.003 | .237±.004 |
| Pred. Stocks | .037±.000 | **.037±.000** | .037±.000 | .037±.000 | .037±.000 | .037±.000 | .037±.000 | .037±.000 | .039±.000 | .037±.000 |
| Pred. Energy | .257±.000 | **.252±.000** | .253±.000 | .253±.000 | .257±.000 | .253±.000 | .256±.000 | .253±.000 | .256±.000 | .253±.000 |

We show t-SNE visualizations and KDE plots for the regular time-series generation in Figure 3 and Figure 1. TimeGAN, GT-GAN, and TimeVAE are representative GAN or VAE-based baselines. In the figures, unlike the baseline methods, the synthetic samples generated from TSGM consistently show successful recall from the original data. Furthermore, TSGM generates diverse synthetic samples in comparison with the three representative baselines in all cases. Especially, TSGM achieves much higher diversity than the baseline models on Air.

## 4.3 SENSITIVITY AND ABLATION STUDIES

### 4.3.1 SENSITIVITY STUDIES

We conduct two sensitivity studies on regular time-series: i) reducing the depth of our score network, ii) decreasing the sampling step numbers. The results are in Table 5.

At first, we modify the depth of our score network from 4 to 3 to check the performance of the lighter conditional score network. Surprisingly, we achieve a better discriminative score with a slight loss on the predictive score.

Next, we decrease the number of sampling steps for faster sampling from 1,000 steps to 500, 250, and 100 steps, respectively. For VP, the case of 500 steps achieves almost the same results as that of original TSGM. Surprisingly, in the case of subVP, we achieve good results until 100 steps.

### 4.3.2 ABLATION STUDIES

Table 6: Comparison between with and without pre-training the autoencoder

| | Method | SDE | Stocks | Energy |
|---|---|---|---|---|
| Disc. | TSGM | VP | .022±.005 | .221±.025 |
| Disc. | TSGM | subVP | **.021±.008** | **.198±.025** |
| Disc. | w/o pre-training | VP | .022±.004 | .322±.003 |
| Disc. | w/o pre-training | subVP | .059±.006 | .284±.004 |
| Pred. | TSGM | VP | .037±.000 | .257±.000 |
| Pred. | TSGM | subVP | **.037±.000** | .252±.000 |
| Pred. | w/o pre-training | VP | .037±.000 | .252±.000 |
| Pred. | w/o pre-training | subVP | .037±.000 | **.251±.000** |

As an ablation study, we simultaneously train the conditional score network, encoder, and decoder from scratch on regular time-series generation (i.e., without the pre-training process). The results are in Table 6. These ablation models are worse than the full model, but they still outperform many baselines. This ablation study shows the efficacy of pre-training our autoencoder. We also provide an additional ablation study in Appendix M.

## 5 CONCLUSIONS

We presented a score-based generative model framework for time-series generation. We combined an autoencoder and our score network into a single framework to accomplish the goal — our framework supports RNN-based or continuous-time method-based autoencoders. We also designed an appropriate denoising score matching loss for our generation task and achieved state-of-the-art results on various datasets in terms of the discriminative and predictive scores. In addition, we conducted rigorous ablation and sensitivity studies to prove the efficacy of our model design.

**Reproducibility.** Our code is available in the supplementary material. We refer the readers to Appendix E, F, G, and J for detail information for reproducibility such as baselines, datasets, hyperparameters, and experimental environments.

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

## A  PROOFS

**Theorem 3.1** (Denoising score matching on time-series). $l_1(n, s)$ can be replaced by the following $l_2(n, s)$:

$$l_2(n, s) = \mathbb{E}_{\mathbf{x}_n^0} \mathbb{E}_{\mathbf{x}_{1:n}^s} \left[ \left\| M_\theta(s, \mathbf{x}_{1:n}^s, \mathbf{x}_{1:n-1}^0) - \nabla_{\mathbf{x}_{1:n}^s} \log p(\mathbf{x}_{1:n}^s | \mathbf{x}_{1:n}^0) \right\|_2^2 \right], \tag{15}$$

where $\mathbf{x}_n^0$ and $\mathbf{x}_{1:n}^s$ are sampled from $p(\mathbf{X}_n^0 | \mathbf{X}_{1:n-1}^0)$ and $p(\mathbf{X}_{1:n}^s | \mathbf{X}_{1:n}^0)$. Therefore, we can use an alternative objective, $L_2 = \mathbb{E}_s \mathbb{E}_{\mathbf{X}_{1:N}} \left[ \sum_{n=1}^N \lambda(s) l_2(n, s) \right]$ instead of $L_1$

*Proof.* At first, if $n = 1$, it can be substituted with the naive denoising score loss by Vincent (2011) since $\mathbf{x}_0^0 = \mathbf{0}$.

Next, let us consider $n > 1$. $l_1(n, s)$ can be decomposed as follows:

$$l_1(n, s) = -2 \cdot \mathbb{E}_{\mathbf{x}_{1:n}^s} \langle M_\theta(s, \mathbf{x}_{1:n}^s, \mathbf{x}_{1:n-1}^0), \nabla_{\mathbf{x}_{1:n}^s} \log p(\mathbf{x}_{1:n}^s | \mathbf{x}_{1:n-1}^0) \rangle$$
$$+ \mathbb{E}_{\mathbf{x}_{1:n}^s} \left[ \left\| M_\theta(s, \mathbf{x}_{1:n}^s, \mathbf{x}_{1:n-1}^0) \right\|_2^2 \right] + C_1 \tag{16}$$

Here, $C_1$ is a constant that does not depend on the parameter $\theta$, and $\langle \cdot, \cdot \rangle$ means the inner product. Then, the first part's expectation of the right-hand side can be expressed as follows:

$$\mathbb{E}_{\mathbf{x}_{1:n}^s} [\langle M_\theta(s, \mathbf{x}_{1:n}^s, \mathbf{x}_{1:n-1}^0), \nabla_{\mathbf{x}_{1:n}^s} \log p(\mathbf{x}_{1:n}^s | \mathbf{x}_{1:n-1}^0) \rangle]$$
$$= \int_{\mathbf{x}_{1:n}^s} \langle M_\theta(s, \mathbf{x}_{1:n}^s, \mathbf{x}_{1:n-1}^0), \nabla_{\mathbf{x}_{1:n}^s} \log p(\mathbf{x}_{1:n}^s | \mathbf{x}_{1:n-1}^0) \rangle p(\mathbf{x}_{1:n}^s | \mathbf{x}_{1:n-1}^0) d\mathbf{x}_{1:n}^s$$
$$= \int_{\mathbf{x}_{1:n}^s} \langle M_\theta(s, \mathbf{x}_{1:n}^s, \mathbf{x}_{1:n-1}^0), \frac{1}{p(\mathbf{x}_{1:n-1}^0)} \frac{\partial p(\mathbf{x}_{1:n}^s, \mathbf{x}_{1:n-1}^0)}{\partial \mathbf{x}_{1:n}^s} \rangle d\mathbf{x}_{1:n}^s$$
$$= \int_{\mathbf{x}_n^0} \int_{\mathbf{x}_{1:n}^s} \langle M_\theta(s, \mathbf{x}_{1:n}^s, \mathbf{x}_{1:n-1}^0), \frac{1}{p(\mathbf{x}_{1:n-1}^0)} \frac{\partial p(\mathbf{x}_{1:n}^s, \mathbf{x}_{1:n-1}^0, \mathbf{x}_n^0)}{\partial \mathbf{x}_{1:n}^s} \rangle d\mathbf{x}_{1:n}^s d\mathbf{x}_n^0$$
$$= \int_{\mathbf{x}_n^0} \int_{\mathbf{x}_{1:n}^s} \langle M_\theta(s, \mathbf{x}_{1:n}^s, \mathbf{x}_{1:n-1}^0), \frac{\partial p(\mathbf{x}_{1:n}^s | \mathbf{x}_{1:n}^0))}{\partial \mathbf{x}_{1:n}^s} \rangle \frac{p(\mathbf{x}_{1:n-1}^0, \mathbf{x}_n^0)}{p(\mathbf{x}_{1:n-1}^0)} d\mathbf{x}_{1:n}^s d\mathbf{x}_n^0 \tag{17}$$
$$= \int_{\mathbf{x}_n^0} \int_{\mathbf{x}_{1:n}^s} \langle M_\theta(s, \mathbf{x}_{1:n}^s, \mathbf{x}_{1:n-1}^0), \frac{\partial p(\mathbf{x}_{1:n}^s | \mathbf{x}_{1:n}^0)}{\partial \mathbf{x}_{1:n}^s} \rangle p(\mathbf{x}_n^0 | \mathbf{x}_{1:n-1}^0) d\mathbf{x}_{1:n}^s d\mathbf{x}_n^0$$
$$= \mathbb{E}_{\mathbf{x}_n^0} \left[ \int_{\mathbf{x}_{1:n}^s} \langle M_\theta(s, \mathbf{x}_{1:n}^s, \mathbf{x}_{1:n-1}^0), \frac{\partial p(\mathbf{x}_{1:n}^s | \mathbf{x}_{1:n}^0)}{\partial \mathbf{x}_{1:n}^s} \rangle d\mathbf{x}_{1:n}^s \right]$$
$$= \mathbb{E}_{\mathbf{x}_n^0} \left[ \int_{\mathbf{x}_{1:n}^s} \langle M_\theta(s, \mathbf{x}_{1:n}^s, \mathbf{x}_{1:n-1}^0), \nabla_{\mathbf{x}_{1:n}^s} \log p(\mathbf{x}_{1:n}^s | \mathbf{x}_{1:n}^0) \rangle p(\mathbf{x}_{1:n}^s | \mathbf{x}_{1:n}^0) d\mathbf{x}_{1:n}^s \right]$$
$$= \mathbb{E}_{\mathbf{x}_n^0} \mathbb{E}_{\mathbf{x}_{1:n}^s} [\langle M_\theta(s, \mathbf{x}_{1:n}^s, \mathbf{x}_{1:n-1}^0), \nabla_{\mathbf{x}_{1:n}^s} \log p(\mathbf{x}_{1:n}^s | \mathbf{x}_{1:n}^0) \rangle]$$

Similarly, the second part's expectation of the right-hand side can be rewritten as follows:

$$
\mathbb{E}_{\mathbf{x}_{1:n}^s}[\|M_\theta(s, \mathbf{x}_{1:n}^s, \mathbf{x}_{1:n-1}^0)\|_2^2]
$$

$$
= \int_{\mathbf{x}_{1:n}^s} \|M_\theta(s, \mathbf{x}_{1:n}^s, \mathbf{x}_{1:n-1}^0)\|_2^2 \cdot p(\mathbf{x}_{1:n}^s|\mathbf{x}_{1:n-1}^0)d\mathbf{x}_{1:n}^s
$$

$$
= \int_{\mathbf{x}_n^0} \int_{\mathbf{x}_{1:n}^s} \|M_\theta(s, \mathbf{x}_{1:n}^s, \mathbf{x}_{1:n-1}^0)\|_2^2 \cdot \frac{p(\mathbf{x}_{1:n}^s, \mathbf{x}_{1:n-1}^0, \mathbf{x}_n^0)}{p(\mathbf{x}_{1:n-1}^0)}d\mathbf{x}_{1:n}^s d\mathbf{x}_n^0
$$

$$
= \int_{\mathbf{x}_n^0} \int_{\mathbf{x}_{1:n}^s} \|M_\theta(s, \mathbf{x}_{1:n}^s, \mathbf{x}_{1:n-1}^0)\|_2^2 \cdot p(\mathbf{x}_{1:n}^s|\mathbf{x}_{1:n}^0)\frac{p(\mathbf{x}_{1:n-1}^0, \mathbf{x}_n^0)}{p(\mathbf{x}_{1:n-1}^0)}d\mathbf{x}_{1:n}^s d\mathbf{x}_n^0 \tag{18}
$$

$$
= \int_{\mathbf{x}_n^0} \int_{\mathbf{x}_{1:n}^s} \|M_\theta(s, \mathbf{x}_{1:n}^s, \mathbf{x}_{1:n-1}^0)\|_2^2 \cdot p(\mathbf{x}_{1:n}^s|\mathbf{x}_{1:n}^0)p(\mathbf{x}_n^0|\mathbf{x}_{1:n-1}^0)d\mathbf{x}_{1:n}^s d\mathbf{x}_n^0
$$

$$
= \mathbb{E}_{\mathbf{x}_n^0}\mathbb{E}_{\mathbf{x}_{1:n}^s}[\|M_\theta(s, \mathbf{x}_{1:n}^s, \mathbf{x}_{1:n-1}^0)\|_2^2]
$$

Finally, by using above results, we can derive following result:

$$
l_1 = \mathbb{E}_{\mathbf{x}_n^0}\mathbb{E}_{\mathbf{x}_{1:n}^s}\left[\|M_\theta(s, \mathbf{x}_{1:n}^s, \mathbf{x}_{1:n-1}^0)\|_2^2\right] + C_1
$$

$$
- 2 \cdot \mathbb{E}_{\mathbf{x}_n^0}\mathbb{E}_{\mathbf{x}_{1:n}^s}\langle M_\theta(s, \mathbf{x}_{1:n}^s, \mathbf{x}_{1:n-1}^0), \nabla_{\mathbf{x}_{1:n}^s}\log p(\mathbf{x}_{1:n}^s|\mathbf{x}_{1:n}^0)\rangle \tag{19}
$$

$$
= \mathbb{E}_{\mathbf{x}_n^0}\mathbb{E}_{\mathbf{x}_{1:n}^s}\left[\|M_\theta(s, \mathbf{x}_{1:n}^s, \mathbf{x}_{1:n-1}^0) - \nabla_{\mathbf{x}_{1:n}^s}\log p(\mathbf{x}_{1:n}^s|\mathbf{x}_{1:n}^0)\|_2^2\right] + C
$$

$C$ is a constant that does not depend on the parameter $\theta$. $\qquad\square$

**Corollary 3.2.** Our target objective function, $L_{score}$, is defined as follows:

$$
L_{score} = \mathbb{E}_s\mathbb{E}_{\mathbf{x}_{1:N}^0}\left[\sum_{n=1}^N \lambda(s)l_2^\star(n, s)\right], \tag{20}
$$

where

$$
l_2^\star(n, s) = \mathbb{E}_{\mathbf{x}_{1:n}^s}\left[\|M_\theta(s, \mathbf{x}_{1:n}^s, \mathbf{x}_{1:n-1}^0) - \nabla_{\mathbf{x}_{1:n}^s}\log p(\mathbf{X}_{1:n}^s|\mathbf{X}_{1:n}^0)\|_2^2\right]. \tag{21}
$$

Then, $L_2 = L_{score}$ is satisfied.

*proof.* Whereas one can use the law of total expectation, which means $E[X] = E[E[X|Y]]$ *if X,Y are on an identical probability space* to show the above formula, we calculate directly. At first, let us simplify the expectation of the inner part with a symbol $f(\mathbf{x}_{1:n}^0)$ for our computational convenience, i.e., $f(\mathbf{x}_{1:n}^0) = \mathbb{E}_s\mathbb{E}_{\mathbf{x}_{1:n}^s}\left[\lambda(s)\|M_\theta(s, \mathbf{x}_{1:n}^s, \mathbf{x}_{1:n-1}^0) - \nabla_{\mathbf{x}_{1:n}^s}\log p(\mathbf{x}_{1:n}^s|\mathbf{x}_{1:n}^0)\|_2^2\right]$. Then we have the following definition:

$$
L_2 = \mathbb{E}_s\mathbb{E}_{\mathbf{x}_{1:N}^0}[l_2] = \mathbb{E}_{\mathbf{x}_{1:N}^0}\left[\sum_{n=1}^N \mathbb{E}_{\mathbf{x}_n^0}[f(\mathbf{x}_{1:n}^0)]\right] = \sum_{n=1}^N \mathbb{E}_{\mathbf{x}_{1:N}^0}\mathbb{E}_{\mathbf{x}_n^0}[f(\mathbf{x}_{1:n}^0)] \tag{22}
$$

At last, the expectation part can be further simplified as follows:

$$\mathbb{E}_{\mathbf{x}_{1:N}^0}\mathbb{E}_{\mathbf{x}_n^0}[f(\mathbf{x}_{1:n}^0)]$$

$$= \int_{\mathbf{x}_{1:N}^0}\int_{\mathbf{x}_n^0} f(\mathbf{x}_{1:n}^0)p(\mathbf{x}_n^0|\mathbf{x}_{1:n-1}^0)d\mathbf{x}_n^0 \cdot p(\mathbf{x}_{1:n-1}^0)p(\mathbf{x}_{n:N}^0|\mathbf{x}_{1:n-1}^0)d\mathbf{x}_{1:N}^0$$

$$= \int_{\mathbf{x}_{1:N}^0}\int_{\mathbf{x}_n^0} f(\mathbf{x}_{1:n}^0)p(\mathbf{x}_{1:n}^0)d\mathbf{x}_n^0 \cdot p(\mathbf{x}_{n:N}^0|\mathbf{x}_{1:n-1}^0)d\mathbf{x}_{1:N}^0$$

$$= \int_{\mathbf{x}_{n:N}^0}\left(\int_{\mathbf{x}_{1:n}^0} f(\mathbf{x}_{1:n}^0)p(\mathbf{x}_{1:n}^0)d\mathbf{x}_{1:n}^0\right) p(\mathbf{x}_{n:N}^0|\mathbf{x}_{1:n-1}^0)d\mathbf{x}_{n:N}^0 \tag{23}$$

$$= \int_{\mathbf{x}_{1:n}^0} f(\mathbf{x}_{1:n}^0)p(\mathbf{x}_{1:n}^0)d\mathbf{x}_{1:n}^0$$

$$= \int_{\mathbf{x}_{1:n}^0}\left(\int_{\mathbf{x}_{n+1:N}^0} p(\mathbf{x}_{n+1:N}^0|\mathbf{x}_{1:n}^0)d\mathbf{x}_{n+1:N}^0\right) f(\mathbf{x}_{1:n}^0)p(\mathbf{x}_{1:n}^0)d\mathbf{x}_{1:n}^0$$

$$= \int_{\mathbf{x}_{1:N}^0} f(\mathbf{x}_{1:n}^0)p(\mathbf{x}_{1:N}^0)d\mathbf{x}_{1:N}^0$$

$$= \mathbb{E}_{\mathbf{x}_{1:N}^0}[f(\mathbf{x}_{1:n}^0)]$$

Since $\sum_{n=1}^N \mathbb{E}_{\mathbf{x}_{1:N}^0}[f(\mathbf{x}_{1:n}^0)] = \mathbb{E}_{\mathbf{x}_{1:N}^0}[\sum_{n=1}^N f(\mathbf{x}_{1:n}^0)] = L_{score}$, we prove the corollary. □

## B    EXISTING TIME-SERIES DIFFUSION MODELS

### B.1    DIFFUSION MODELS FOR TIME-SERIES FORECASTING AND IMPUTATION

TimeGrad (Rasul et al., 2021) is a diffusion-based method for time-series forecasting, and CSDI (Tashiro et al., 2021) is for time-series imputation.

In TimeGrad (Rasul et al., 2021), they used a diffusion model for forecasting future observations given past observations. On each sequential order $n \in \{2, ..., N\}$ and diffusion step $s \in \{1, ..., T\}$, they train a neural network $\epsilon_\theta(\cdot, \mathbf{x}_{1:n-1}, s)$ with a time-dependent diffusion coefficient $\bar{\alpha}_s$ by minimizing the following objective function:

$$\mathbb{E}_{\mathbf{x}_n^0,\epsilon,s}[\||\epsilon - \epsilon_\theta(\sqrt{\bar{\alpha}_s}\mathbf{x}_n^0 + \sqrt{1-\bar{\alpha}_s}\epsilon, \mathbf{x}_{1:n-1}, s)\||_2^2], \tag{24}$$

where $\epsilon \sim \mathcal{N}(\mathbf{0}, \boldsymbol{I})$. The above formula assumes that we already know $\mathbf{x}_{1:n-1}$, and by using an RNN encoder, $\mathbf{x}_{1:n-1}$ can be encoded into $\mathbf{h}_{n-1}$. After training, the model forecasts future observations recursively. More precisely speaking, $\mathbf{x}_{1:n-1}$ is encoded into $\mathbf{h}_{n-1}$ and the next observation $\mathbf{x}_n$ is forecast from the previous condition $\mathbf{h}_{n-1}$.

CSDI (Tashiro et al., 2021) proposed a general diffusion framework which can be applied mainly to time-series imputation. CSDI reconstructs an entire sequence at once, not recursively. Let $\mathbf{x}^0 \in \mathbb{R}^{\dim(\mathbf{X}) \times N}$ be an entire time-series sequence with $N$ observations in a matrix form. They define $\mathbf{x}_{co}^0$ and $\mathbf{x}_{ta}^0$ as conditions and imputation targets which are derived from $\mathbf{x}^0$, respectively. They then train a neural network $\epsilon_\theta(\cdot, \mathbf{x}_{co}^0, s)$ with a corresponding diffusion coefficient $\bar{\alpha}_s$ and a diffusion step $s \in \{1, ..., T\}$ by minimizing the following objective function:

$$\mathbb{E}_{\mathbf{x}^0,\epsilon,s}[\||\epsilon - \epsilon_\theta(s, \mathbf{x}_{ta}^s, \mathbf{x}_{co}^0)\||_2^2], \tag{25}$$

where $\mathbf{x}_{ta}^s = \sqrt{\bar{\alpha}_s}\mathbf{x}_{ta}^0 + (1-\bar{\alpha}_s)\epsilon$. By training the network using the above loss, it generates missing elements from the partially filled matrix $\mathbf{x}_{co}^0$.

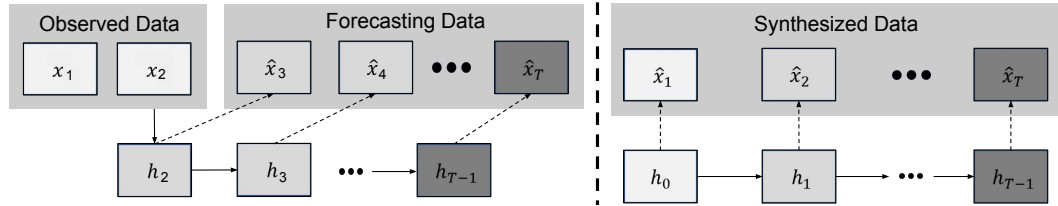

Figure 4: Graphical representation of TimeGrad (left) and TSGM (right). We adapt TimeGrad to our generation task but its results are not comparable even to other baselines' results (see Appendix C.1).

## B.2 DIFFERENCE BETWEEN EXISTING AND OUR WORKS

Although they have earned state-of-the-art results for forecasting and imputation, we found that they are not suitable for our generative task due to the fundamental mismatch between their model designs and our task (cf. Table 7 and Fig. 4).

Table 7: Comparison among various recent GAN, diffusion, and SGM-based methods for time-series. $\mathbf{x}_t$ (resp. $\hat{\mathbf{x}}_t$) means a raw (resp. synthesized) observation at time $t$. For CSDI, $\mathbf{x}_{co}$ means a set of known values and $\mathbf{x}_{ta}$ means a set of target missing values — it is not necessary that $\mathbf{x}_{co}$ precedes $\mathbf{x}_{ta}$ in time in CSDI.

| Method | Type | Task Description |
|--------|------|------------------|
| TimeGrad | Diffusion | From $\mathbf{x}_{1:N-K}$, infer $\hat{\mathbf{x}}_{N-K+1:N}$. |
| CSDI | Diffusion | Given known values $\mathbf{x}_{co}$, infer missing values $\hat{\mathbf{x}}_{ta}$. |
| TimeGAN | GAN | Synthesize $\hat{\mathbf{x}}_{1:N}$ from scratch. |
| GT-GAN | GAN | Synthesize $\hat{\mathbf{x}}_{1:N}$ from scratch. |
| TSGM | SGM | Synthesize $\hat{\mathbf{x}}_{1:N}$ from scratch. |

TimeGrad generates future observations given the hidden representation of past observations $\mathbf{h}_{n-1}$, i.e., a typical forecasting problem. Since our task is to synthesize from scratch, past known observations are not available. Thus, TimeGrad cannot be directly applied to our task.

In CSDI, there are no fixed temporal dependencies between $\mathbf{x}_{co}^0$ and $\mathbf{x}_{ta}^0$ since its task is to impute missing values, i.e., $\mathbf{x}_{ta}^0$, from known values, i.e., $\mathbf{x}_{co}^0$, in the matrix $\mathbf{x}^0$. It is not necessary that $\mathbf{x}_{co}^0$ precedes $\mathbf{x}_{ta}^0$ in time, according to the CSDI's method design. Our synthesis task can be considered as $\mathbf{x}_{co}^0 = \emptyset$, which is the most extreme case of the CSDI's task. Therefore, it is not suitable to be used for our task.

To our knowledge, we are the first proposing an SGM-based time-series synthesis method. We propose to train a conditional score network by using the denoising score matching loss proposed by us, which is denoted as $L_{score}^{\mathcal{H}}$. Unlike other methods (Rasul et al., 2021; Tashiro et al., 2021) that resort to existing known proofs, we design our denoising score matching loss in Eq. equation 12 and prove its correctness. Meanwhile, TimeGrad and CSDI can be somehow modified for time-series synthesis but their generation quality is mediocre (see Appendix C).

## C EXPERIMENTAL RESULTS FOR INAPPLICABILITY OF EXISTING TIME-SERIES DIFFUSION MODELS TO OUR WORK

In this section, we provide experimental results to show inapplicability of the existing time-series diffusion models, TimeGrad and CSDI, to the time-series generation task.

### C.1 ADAPTING TIMEGRAD TOWARD GENERATION TASK

In this section, TimeGrad (Rasul et al., 2021) is modified for the generation task. We simply add an artificial zero vector $\mathbf{0}$ in front of the all time-series samples of Energy. Therefore, TimeGrad's task becomes given a zero vector, forecasting (or generating) all other remaining observations. For the stochastic nature of its forecasting process, it can somehow generate various next observations given

Table 8: Comparison between TSGM and modified TimeGrad in Energy for its regular time-series setting

| Method | Disc. | Pred. |
|---|---|---|
| TSGM-VP | .221±.025 | .257±.000 |
| TSGM-subVP | .198±.025 | .252±.000 |
| Modified TimeGrad | .500±.000 | .287±.003 |

Table 9: Comparison between TSGM and modified CSDI in Energy and AI4I for its regular time-series setting

| Method | Energy | | AI4I | |
|---|---|---|---|---|
| | Disc. | Pred. | Disc. | Pred. |
| TSGM-VP | .221±.025 | .257±.000 | .147±.005 | .217±.000 |
| TSGM-subVP | .198±.025 | .252±.000 | .150±.010 | .217±.000 |
| Modified CSDI | .500±.000 | .641±.000 | .500±.000 | .640±.000 |

the sample input $\mathbf{0}$. Table 8 shows the experimental comparison between modified TimeGrad and TSGM in Energy, which has high dimensional features, for its regular time-series setting. TSGM gives outstanding performance, compared to modified TimeGrad. When checked in Table 3, modified TimeGrad is even worse than some baselines. Therefore, unlike TSGM, TimeGrad is not appropriate for the generation task.

### C.2 ADAPTING CSDI TOWARD GENERATION TASK

In this section, we apply CSDI to the time-series generation task by regarding all observations as missing values (i.e., $\mathbf{x}_{co}^0 = \mathbf{0}$). However, as demonstrated in Table 9, CSDI fails to generate reliable time series samples in the Energy and AI4I datasets for its regular time series setting. Hence, we conclude that CSDI is unsuitable for the time-series generation task.

## D DETAILED TRAINING PROCEDURE

We train the conditional score network and the encoder-decoder pair alternately after the pre-training step. For some datasets, we found that training only the conditional score network achieves better results after pre-training the autoencoder. Therefore, $use_{alt} = \{True, False\}$ is a hyperparameter to set whether we use the alternating training method. We give the detailed training procedure in Algorithm 1.

---

**Algorithm 1:** Training algorithm

**Input:** $\mathbf{x}_{1:N}^0$; $use_{alt}$ is a Boolean parameter to set whether to use the alternating training method; $iter_{pre}$ is the number of iterations for pre-training; $iter_{main}$ is the number of iterations for training.

1 **for** $iter \in \{1, ..., iter_{pre}\}$ **do**
2     Train $Encoder$ and $Decoder$ by using $L_{ed}$
3 **end**
4 **for** $iter \in \{1, ..., iter_{main}\}$ **do**
5     Train $M_\theta$ by using $L_{score}^{\mathcal{H}}$
6     **if** $use_{alt}$ **then**
7         Train the $Encoder$ and $Decoder$ by using $L_{ed}$
8     **end**
9 **end**
10 **return** $Encoder, Decoder, M_\theta$

---

## E DATASETS AND BASELINES

We use 4 datasets from various fields as follows. We summarize their data dimensions, the number of training samples, and their time-series lengths (window sizes) in Table 10.

- *Stock* (Yoon et al., 2019): The Google stock dataset was collected irregularly from 2004 to 2019. Each observation has (volume, high, low, opening, closing, adjusted closing prices), and these features are correlated.

- *Energy* (Candanedo et al., 2017): This dataset is from the UCI machine learning repository for predicting the energy use of appliances from highly correlated variables such as house temperature and humidity conditions.

- *Air* (De Vito et al., 2008): The UCI Air Quality dataset was collected from 2004 to 2005. Hourly averaged air quality records are gathered using gas sensor devices in an Italian city.

- *AI4I* (Matzka, 2020): AI4I means the UCI AI4I 2020 Predictive Maintenance dataset. This data reflects the industrial predictive maintenance scenario with correlated features including several physical quantities.

We use several types of generative methods for time-series as baselines. At first, we consider autoregressive generative methods: T-Forcing (teacher forcing) (Graves, 2013; Sutskever et al., 2011) and P-Forcing (professor forcing) (Goyal et al., 2016). Next, we use GAN-based methods: TimeGAN (Yoon et al., 2019), RCGAN (Esteban et al., 2017), C-RNN-GAN (Mogren, 2016), COT-GAN (Xu et al., 2020), GT-GAN (Jeon et al., 2022). We also test VAE-based methods into our baselines: TimeVAE (Desai et al., 2021). Finally, we treat flow-based methods. Among the array of flow-based models designed for time series generation, we have chosen to compare our TSGM against CTFP (Deng et al., 2020). This choice is informed by the fact that CTFP possesses the capability to handle both regular and irregular time series samples, aligning well with the nature of our task which involves generating both regular and irregular time series data.

Table 10: Characteristics of the datasets we use for our experiments

| Dataset | Dimension | #Samples | Length |
|---------|-----------|----------|--------|
| Stocks | 6 | 3685 | |
| Energy | 28 | 19735 | 24 |
| Air | 13 | 9357 | |
| AI4I | 5 | 10000 | |

## F  Hyperparameters and its Search Space

Table 11 shows the best hyperparameters for our conditional score network $M_\theta$ on regular time-series, and we explain its neural network architecture in Appendix J.2. $M_\theta$ has various hyperparameters and for key hyperparameters, we set them as listed in Table 11. For other common hyperparameters with baselines, we reuse the default configurations of TimeGAN (Yoon et al., 2019) and VPSDE (Song et al., 2021) to conduct the regular time-series generation.

We give our search space for the hyperparameters of TSGM. $iter_{pre}$ is in {50000,100000}. The dimension of hidden features, $d_{hidden}$, ranges from 2 times to 5 times the dimension of input features. On regular time-series generation, we follow the default values in TimeGAN (Yoon et al., 2019) and VPSDE (Song et al., 2021). For irregular time-series tasks, we search the hidden dimension of decoder from 2 times to 4 times the dimension of input dimension, and follow GTGAN (Jeon et al., 2022) for other settings of NCDE-encoder and GRU-ODE-decoder. We give our best hyperparameters for irregular time-series on Table 12.

For baselines, we check their hyperparameters as follow:

- T-forcing (Graves, 2013): We control batch size among {256, 512, 1024}.

- P-forcing (Goyal et al., 2016): We control batch size among {256, 512, 1024}.

- TimeGAN (Yoon et al., 2019): The dimension of hidden features range from 2 times to 4 times the dimension of input features.

- RCGAN (Esteban et al., 2017): We control learning rate of generator's optimizer and discriminator's optimizer from {1e-4, 2e-4} and {1e-3, 5e-3}, respectively.

- C-RNN-GAN (Mogren, 2016): We control learning rate of generator's optimizer and discriminator's optimizer from {1e-4, 2e-4} and {3e-4, 4e-4}, respectively. We also use label smoothing which is stated in the paper.
- TimeVAE (Desai et al., 2021): We control its latent dimension among {5, 10, 20}.
- COT-GAN (Xu et al., 2020): We calculate score every 250 epoch during 1000 epochs and get the best experimental results.
- CTFP (Deng et al., 2020): The dimension of hidden features range from 2 times to 4 times the dimension of input features.
- GT-GAN (Jeon et al., 2022): For encoder-decoder pair, we test from exactly the same search space as TSGM. We calculate score every 5000 iteration during 40000 iterations and get the best score.

Especially for COT-GAN, since it is on video generation, modifying the architecture to one dimensional form was difficult. So, we augment our time-series data into two dimensional ones by stacking them. After generating two-dimensional data, we extract the first row of the synthesized one and calculate the score. We search every hyperparameter from {0.5, 1, 2} times of default value. Through the experiment, we acquire compatible result but lower than TimeGAN in several datasets.

We follow default values for miscellaneous settings which are not explained on the above. Additionally, to deal with irregular time-series, we search the hyperparameters of GRU-D, which substitutes for RNN or are added to the head of baselines. We test the hidden dimension of GRU-D from 2 times to 4 times the dimension of input features.

Table 11: The best hyperparameter setting for our method on regular time-series.

| Dataset | $\dim(\mathbf{h})$ | $use_{alt}$ | $iter_{pre}$ | $iter_{main}$ |
|---------|------|------|--------|--------|
| Stocks | 24 | True | 50000 | |
| Energy | 56 | False | 100000 | 40000 |
| Air | 40 | True | 50000 | |
| AI4I | 24 | True | 50000 | |

Table 12: The best hyperparameter setting for our method on irregular time-series. $D_{hidden}$ denotes the hidden dimension of GRU-ODE-decoder.

| Dataset | $D_{hidden}$ | $\dim(\mathbf{h})$ | $use_{alt}$ | $iter_{pre}$ | $iter_{main}$ |
|---------|---------|------|------|--------|--------|
| Stocks | 48 | 24 | True | | |
| Energy | 112 | 56 | False | 50000 | 40000 |
| Air | 40 | 40 | True | | |
| AI4I | 48 | 24 | True | | |

## G  MISCELLANEOUS EXPERIMENTAL ENVIRONMENTS

We give detailed experimental environments. The following software and hardware environments were used for all experiments: UBUNTU 18.04 LTS, PYTHON 3.9.12, CUDA 9.1, NVIDIA Driver 470.141, i9 CPU, and GEFORCE RTX 2080 TI.

In the experiments, we report only the VP and subVP-based TSGM and exclude the VE-based one for its lower performance. For baselines, we reuse their released source codes in their official repositories and rely on their designed training and model selection procedures. For our method, we select the best model for every 5000 iterations. For this, we synthesize samples and calculate the mean and standard deviation scores of the discriminative and predictive scores.

## H  EMPIRICAL SPACE AND TIME COMPLEXITY ANALYSES

We report the memory usage during training in Table 13 and the wall-clock time for generating 1,000 time-series samples in Table 14. We compare TSGM to TimeGAN (Yoon et al., 2019) and GTGAN (Jeon et al., 2022). Our method is relatively slower than TimeGAN and GTGAN, which

is a fundamental drawback of all SGMs. For example, the original score-based model (Song et al., 2021) requires 3,214 seconds for sampling 1,000 CIFAR-10 images while StyleGAN (Karras et al., 2019) needs 0.4 seconds. However, we also emphasize that this problem can be relieved by using the techniques suggested in (Xiao et al., 2022; Jolicoeur-Martineau et al., 2021) as we mentioned in the conclusion section.

Table 13: The memory usage for training

| Method | Stock | Energy |
|--------|-------|--------|
| TimeGAN | 1.1 (GB) | 1.6 (GB) |
| GTGAN | 2.3 (GB) | 2.3 (GB) |
| TSGM | 3.8 (GB) | 3.9 (GB) |

Table 14: The sampling time of TSGM, TimeGAN and GTGAN for generating 1,000 samples on each dataset. The original score-based model (Song et al., 2021) requires 3,214 seconds for sampling 1000 CIFAR-10 images while StyleGAN (Karras et al., 2019) needs 0.4 seconds, which is similar to the case between TSGM and TimeGAN.

| Method | Stocks | Energy |
|--------|--------|--------|
| TimeGAN | 0.43 (s) | 0.47 (s) |
| GTGAN | 0.43 (s) | 0.47 (s) |
| TSGM | 3318.99 (s) | 1620.84 (s) |

Although our method achieves state-of-the-art sampling quality and diversity, there exists a fundamental problem that all SGMs have. That is, SGMs are slower than GANs for generating samples (see above). Since there are several accomplishments for faster sampling (Xiao et al., 2022; Jolicoeur-Martineau et al., 2021), however, one can apply them to our method and it would be much faster without any loss of sampling quality and diversity.

## I   ENCODER AND DECODER FOR IRREGULAR TIME-SERIES

To process irregular time-series, one can use continuous-time methods for constructing the encoder and the decoder. In our case, we use neural controlled differential equations (NCDEs) for designing the encoder and GRU-ODEs for designing the decoder, respectively (Kidger et al., 2020; Brouwer et al., 2019). Our encoder based on NCDEs can be defined as follows:

$$\mathbf{h}(t_n) = \mathbf{h}(t_{n-1}) + \int_{t_{n-1}}^{t_n} f(t, \mathbf{h}(t); \theta_f) \frac{dX(t)}{dt} dt, \tag{26}$$

where $X(t)$ is an interpolated continuous path from $\mathbf{x}_{1:N}$ — NCDEs typically use the natural cubic spline algorithm to define $X(t)$, which is twice differentiable and therefore, there is not any problem to be used for forward inference and backward training. In other words, NCDEs evolve the hidden state $\mathbf{h}(t)$ by solving the above Riemann-Stieltjes integral.

For the decoder, one can use the following GRU-ODE-based definition:

$$\overline{\mathbf{d}}(t_n) = \mathbf{d}(t_{n-1}) + \int_{t_{n-1}}^{t_n} g(t, \mathbf{d}(t); \theta_g) dt, \qquad \mathbf{d}(t_n) = \text{GRU}(\mathbf{h}(t_n), \overline{\mathbf{d}}(t_n)), \qquad \hat{\mathbf{x}}_n = FC(\mathbf{d}(t_n)), \tag{27}$$

where $FC$ denotes a fully-connected layer-based output layer. The intermediate hidden representation $\overline{\mathbf{d}}(t_n)$ is jumped into the hidden representation $\mathbf{d}(t_n)$ by the GRU-based jump layer. At the end, there is an output layer.

For our irregular time-series experiments, i.e, dropping 30%, 50%, and 70% of observations from regular time-series, we use the above encoder and decoder definitions and have good results.

## J  NEURAL NETWORK ARCHITECTURE

### J.1  ARCHITECTURAL DETAILS OF NCDES AND GRU-ODES

As mentioned in Appendix I, we take the following architecture for functions $f$, $g$ of (26) and (27) in Table 15.

Table 15: Architecture of functions $f$(upper) and $g$(lower). Each layer of encoder and gate of decoder takes ($\sigma \circ \text{Linear}$) form where $\sigma$ denotes activation function. We describe which activation and Linear function are used.

| Layer | Activation function | Linear |
|:-----:|:-------------------:|:------:|
| 1 | ReLU | $\dim(\mathbf{x}) \to 4\dim(\mathbf{x})$ |
| 2 | ReLU | $4\dim(\mathbf{x}) \to 4\dim(\mathbf{x})$ |
| 3 | ReLU | $4\dim(\mathbf{x}) \to 4\dim(\mathbf{x})$ |
| 4 | Tanh | $4\dim(\mathbf{x}) \to \dim(\mathbf{x})$ |

| Layer | Gate | Activation function | Linear |
|:-----:|:----:|:-------------------:|:------:|
| | $r_t$ | ReLU | |
| 1 | $z_t$ | ReLU | $\dim(\mathbf{h}) \to \dim(\mathbf{h})$ |
| | $u_t$ | Tanh | |

### J.2  CONDITIONAL SCORE NETWORK

Unlike other generation tasks, e.g., image generation (Song et al., 2021) and tabular data synthesis (Kim et al., 2022), where each sample is independent, time-series observations are dependent to their past observations. Therefore, the score network for time-series generation must be designed to learn the conditional log-likelihood given past generated observations, which is more complicated than that in image generation.

In order to learn the conditional log-likelihood, we modify the popular U-net (Ronneberger et al., 2015) architecture for our purposes. Since U-net has achieved various excellent results for other generative tasks (Song & Ermon, 2019; Song et al., 2021), we modify its 2-dimensional convolution layers to 1-dimensional ones for handling time-series observations. The modified U-net, denoted $M_\theta$, is trained to learn our conditional score function (cf. Eq. equation 12). More details on training and sampling with $M_\theta$ are in Sec. 3.4.

## K  ADDITIONAL EXPERIMENTAL RESULTS

We give additional experimental results for irregular time-series generation with 50% and 70% missing rates in Table 16.

## L  ADDITIONAL VISUALIZATIONS

In this section, we provide additional visualization results in each dataset. Figure 5 illustrates the density function of each feature estimated by KDE in original and generated data. Figure 6 shows original and generated data points projected onto a latent space using t-SNE (van der Maaten & Hinton, 2008)

## M  EFFICACY OF OUR RECURSIVE GENERATION

In this section, we investigate the efficacy of our proposed recursive design. We compare TSGM to an method using *one-shot generation*. we call *one-shot generation* when a generation method generates all time-series observations at once, not recursively. In other words, $D \times N$ matrices, where $D$ means the number of features and $N$ means the sequence length, are synthesized at once. CSDI (Tashiro et al., 2021) is one of the most famous one-shot imputation model for time-series.

Table 16: Experimental results in terms of the discriminative and predictive scores. The best scores are in boldface. The left and right ones denote experimental results on irregular time-series with 50% and 70% missing rates, respectively.

| | Method | Stocks | Energy | Air | AI4I | Method | Stocks | Energy | Air | AI4I |
|---|---|---|---|---|---|---|---|---|---|---|
| Disc. score | TSGM-VP | .051±.014 | .398±.003 | .272±.012 | .156±.106 | TSGM-VP | .065±.010 | .482±.003 | .337±.025 | .327±.104 |
| | TSGM-subVP | **.031±.012** | .421±.008 | **.213±.025** | **.137±.102** | TSGM-subVP | **.035±.009** | **.213±.025** | **.329±.027** | **.235±.123** |
| | T-Forcing-D | .407±.034 | .376±.046 | .499±.001 | .473±.045 | T-Forcing-D | .404±.068 | .336±.032 | .499±.001 | .493±.010 |
| | P-Forcing-D | .500±.000 | .500±.000 | .494±.012 | .437±.079 | P-Forcing-D | .449±.150 | .494±.011 | .498±.002 | .440±.125 |
| | TimeGAN-D | .477±.021 | .473±.015 | .500±.001 | .500±.000 | TimeGAN-D | .485±.022 | .500±.000 | .500±.000 | .500±.000 |
| | RCGAN-D | .500±.000 | .500±.000 | .500±.000 | .500±.000 | RCGAN-D | .500±.000 | .500±.000 | .500±.000 | .500±.000 |
| | C-RNN-GAN-D | .500±.000 | .500±.000 | .500±.000 | .450±.150 | C-RNN-GAN-D | .500±.000 | .500±.000 | .500±.000 | .500±.000 |
| | TimeVAE-D | .411±.110 | .436±.088 | .423±.153 | .389±.113 | TimeVAE-D | .444±.148 | .498±.003 | .426±.148 | .371±.092 |
| | COT-GAN-D | .499±.001 | .500±.000 | .500±.000 | .500±.000 | COT-GAN-D | .498±.001 | .500±.000 | .500±.000 | .500±.000 |
| | CTFP | .499±.000 | .500±.000 | .500±.000 | .499±.001 | CTFP | .500±.000 | .500±.000 | .500±.000 | .499±.000 |
| | GT-GAN | .265±.073 | **.317±.010** | .434±.035 | .276±.033 | GT-GAN | .230±.053 | .325±.047 | .444±.019 | .362±.043 |
| Pred. score | TSGM-VP | .011±.000 | .051±.001 | **.041±.001** | **.060±.001** | TSGM-VP | **.011±.000** | .053±.001 | .043±.000 | .092±.024 |
| | TSGM-subVP | **.011±.000** | **.051±.001** | .042±.002 | .065±.013 | TSGM-subVP | .012±.000 | **.042±.002** | **.042±.001** | **.097±.020** |
| | T-Forcing-D | .038±.003 | .090±.000 | .121±.003 | .143±.005 | T-Forcing-D | .031±.002 | .091±.000 | .116±.003 | .144±.004 |
| | P-Forcing-D | .089±.010 | .198±.005 | .101±.003 | .116±.007 | P-Forcing-D | .107±.009 | .193±.006 | .107±.002 | .125±.007 |
| | TimeGAN-D | .254±.047 | .339±.029 | .325±.005 | .251±.010 | TimeGAN-D | .228±.000 | .443±.000 | .425±.008 | .323±.011 |
| | RCGAN-D | .333±.044 | .250±.010 | .335±.023 | .276±.066 | RCGAN-D | .441±.045 | .349±.027 | .359±.008 | .346±.029 |
| | C-RNN-GAN-D | .273±.000 | .438±.000 | .289±.033 | .373±.037 | C-RNN-GAN-D | .281±.019 | .436±.000 | .306±.040 | .262±.053 |
| | TimeVAE-D | .195±.012 | .143±.007 | .103±.002 | .144±.004 | TimeVAE-D | .199±.009 | .134±.004 | .108±.004 | .142±.008 |
| | COT-GAN-D | .246±.000 | .475±.000 | .557±.000 | .449±.000 | COT-GAN-D | .278±.000 | .456±.000 | .556±.000 | .435±.000 |
| | CTFP | .084±.005 | .469±.008 | .476±.235 | .412±.024 | CTFP | .084±.005 | .469±.008 | .476±.235 | .412±.024 |
| | GT-GAN | .018±.002 | .064±.001 | .061±.003 | .113±.024 | GT-GAN | .020±.005 | .076±.001 | .059±.004 | .124±.003 |
| | Original | .011±.002 | .045±.001 | .044±.006 | .059±.001 | Original | .011±.002 | .045±.001 | .044±.006 | .059±.001 |

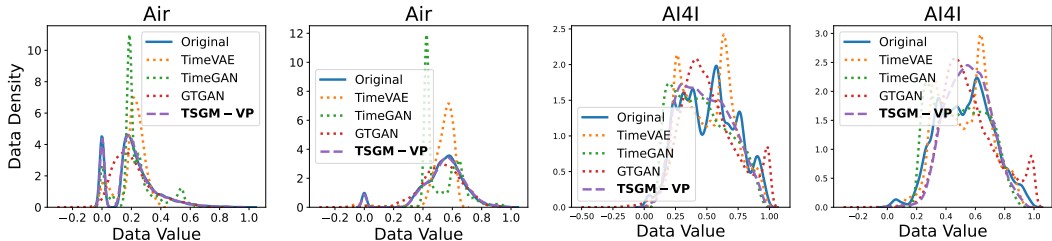

Figure 5: Additional KDE plots for each feature in Air and AI4I datasets.

We convert our TSGM for the one-shot generation by removing the RNN-based encoder. In Table 17, TSGM-oneshot shows poor generation quality in Stock and Energy. TSGM-oneshot achieves comparable predictive scores but its discriminative score gets worse a lot. From these results, we can support the efficacy of our recursive structures, compared to one-shot generation. One can also check the one-shot generation result by CSDI in Appendix C.2.

# N    DRIFT AND DIFFUSION TERMS IN VE, VP AND SUBVP SDE

In this section, we describe the detailed form of each SDE. In (Song et al., 2021), the authors investigated that SMLD (Song & Ermon, 2019) and DDPM (Ho et al., 2020) can be extended to

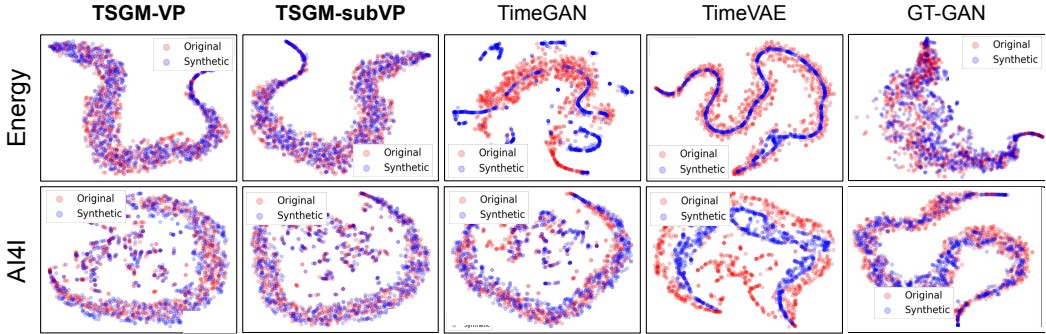

Figure 6: Additional t-SNE plots in Energy and AI4I datasets.

Table 17: Comparison between TSGM and one-shot generations. We give representative results. For other datasets, the results are similar or worse than the table.

| Method | Stock | | Energy | |
|---|---|---|---|---|
| | Disc. | Pred. | Disc. | Pred. |
| TSGM-VP | .022±.005 | .037±.000 | .221±.025 | .257±.000 |
| TSGM-subVP | .021±.008 | .037±.000 | .198±.025 | .252±.000 |
| TSGM-oneshot | .029±.018 | .037±.000 | .494±.001 | .258±.000 |

continuous forms and as a result, suggested VE and VP SDEs. Furthermore, the author proposed an additional SDE form, called subVP SDE, which has a smaller variance than VP SDE but the same expectation. The exact calculation is not a main subject of this paper, so we only explain the form of these terms in Table 2. Please refer to (Song et al., 2021) for the detailed computation.

Along with already mentioned notations in Section 2.1, we define noise scales. $\sigma(s)$ means positive noise values which are increasing, and $\beta(s)$ denotes noise values in [0,1] which are used in SMLD and DDPM. Although we give the exact form of the three SDEs, we report only the VP and subVP-based TSGM in our experiments and exclude the VE-based one for its lower performance.

## O    DETAILED DESCRIPTION OF GRU-D

GRU-D (Che et al., 2016) is a modified GRU model which is for learning time-series data with missing values. This concept is similar with our problem statement, so we apply it to our baseline for irregular case. GRU-D needs to learn decaying rates along with the values of GRU. First, GRU-D learns decay rates which depict vagueness of data as time passed. After calculating the decay rates, each value is composed of decay rate, mask, latest observed data, and predicted empirical mean that of GRU. The code can be utilized in the following link: https://github.com/zhiyongc/GRU-D

