# OpenReview forum: "TSGM: Regular and Irregular Time-series Generation using Score-based Generative Models"
_ICLR.cc/2024/Conference — Submitted to ICLR 2024_

### Official Review · Reviewer_ox1H · 2023-11-01

**Soundness:** 2 fair
**Presentation:** 2 fair
**Contribution:** 2 fair
**Rating:** 5
**Confidence:** 3

**Summary:**

In this paper, the authors propose a score-based generative models, namely TSGM, that can synthetic both regular time series and irregular time series. The novelty of the paper lies in the loss for score network. The authors conduct comprehensive experiments to demonstrate the effectiveness of the method and the authors also acknowledge the high run-time issue.

**Strengths:**

1. The authors propose the first SGM-based method for synthetic both regular time series and irregular time series.

2. The authors propose a novel loss function for score network on time-series data.

3. The proposed model achieves state-of-the-art results on different settings.

**Weaknesses:**

1. The propose loss function is not clearly written.

2. The time complexity of TSGM is not discussed.

3. The procedure of generating irregular time series is not clearly written.
(See questions for more details.)

**Questions:**

1. What is the time complexity of TSGM compared to baseline.

2. What is the diffusion procedure for generating irregular time series using continuous-time models as encoder?

3. As for irregular time series generation, how is the irregular-sampled timestamp generated in your sampling procedure?

4. Please make a more clear explanation of how Eq. (12) is calculated.

5. How does the loss function related to time series data generation?

6. How does the proposed loss function in Eq. (12) solve the computation issue? Does it accelerate the training process or the sampling procedure?

7. Which encoder are used for encoding the irregular time series in your experiment?

---

> ### Author Response · Authors · 2023-11-16
>
> **Q1.What is the time complexity of TSGM compared to baseline?**
>
> We compared the sampling time of TSGM to TimeGAN and GTGAN in Appendix H. Please see Table 13.
>
> **Q2.What is the diffusion procedure for generating irregular time series using continuous-time models as encoder?**
>
> It is conceptually the same at the one for regular time-series. The only difference is whether its time interval is fixed or irregular. It is well-known that Neural CDEs are continuous RNNs [1]. So, we use Neural CDEs to deal with irregular time-series. Therefore, we train our score network by using latent variables, which is encoded by our Neural CDE-based encoder, and generate recursively latent variables as we have done in the regular time-series generation.
>
> [1] Patrick Kidger. Neural Controlled Differential Equations for Irregular Time Series. NeurIPS20.
>
> **Q3.As for irregular time series generation, how is the irregular-sampled timestamp generated in your sampling procedure?**
>
> We randomly drop 30\%, 50\%, and 70\% of observations from regular time-series. We followed the procedure of how to make irregular-sampling timestamps by GTGAN for fair comparison. We add a description of the procedure to the problem formulation section, Section 3.1.
>
> **Q4.Please make a more clear explanation of how Eq. (12) is calculated.
> Q5.How does the loss function related to time series data generation?**
>
> Unlike image generation, time-series must be generated by considering past observations. But previous researches did not consider such a conditional generation, so we had to derive objective for conditional score function, ${\nabla}log p({\textbf{x}}^s_{1:n}|\textbf{x}^0_{1:n-1})$.
>
> But calculating an exact value of ${\nabla}log p({\textbf{x}}^s_{1:n}|\textbf{x}^0_{1:n-1})$ is computationally prohibitive [1]. As ${\nabla}log p({\textbf{x}}^s_{1:n}|\textbf{x}^0_{1:n})$ has a definite form and follows a specific SDE (e.g., VE, VP, subVP) [2], we drive that $L_1$ is same with $L_{score}$ by constant term.
>
> Considering that using $\textbf{x}_{1:n}$ is computationally expensive, we use an autoencoder to encode it to latent variable.
>
> Once we generate $\textbf{h}_{1:N}$, we acquire total sequence by using a trained decoder.
>
> As a result, our loss function for the score network is $L_{score}^{\mathcal{H}}$.
>
> [1] Aapo Hyv{{\"a}}rinen. Estimation of Non-Normalized Statistical Models by Score Matching. Journal of Machine Learning Research, 2005.
>
> [2] Simo Särkkä. Applied stochastic differential equations, volume 10. Cambridge University Press, 2019.
>
> **Q6.How does the proposed loss function in Eq. (12) solve the computation issue? Does it accelerate the training process or the sampling procedure?**
>
> As we mentioned above, calculating ${\nabla}log p({\textbf{x}}^s_{1:n}|\textbf{x}^0_{1:n-1})$ is computationally prohibitive. So we derive that train ${\nabla}log p({\textbf{x}}^s_{1:n}|\textbf{x}^0_{1:n})$ is the same as in [1]. It is not related to accelerating, but training.
>
> [1] Vincent. A connection between score matching and denoising autoencoders. Neural Computation, 23(7):1661–1674, 2011.
>
> **Q7.Which encoder are used for encoding the irregular time series in your experiment?**
>
> As we explained Q2, we use Neural CDE for irregular time-series. The detailed architecture is described in Appendix I and J.

---

> > ### Author Response · Authors · 2023-11-22
> >
> > Dear Reviewer ox1H,
> >
> > We treat your whole comments with appropriate answers.
> >
> > Could you please check our rebuttal above and let us know whether you have more concerns?
> >
> > Best, Authors.

---

> > ### Comment · Reviewer_ox1H · 2023-11-22
> > **Thank you for your prompt response.**
> >
> > > Q1. What is the time complexity of TSGM compared to baseline?
> >
> > Q1.1: I mean the formal time complexity analysis of these methods. Could you please give any formal analysis? What is the bottleneck of this method?
> >
> > Q1.2: I also wonder if there is any method through which we could reduce the time complexity or computational cost. I have noticed the results you reported in Table 13, the time cost is significantly worse than the compared baselines. The major weakness of this method is the computational cost and time complexity. Considering the cost this this method, it is impractical for this method in real-world scenarios. The authors should pay more attention to this part.
> >
> > Besides, the author should also claim the motivation of specifically utilizing SGMs (score-based generation models) for time series generation. In the original paper, the author claimed that "Despite the previous efforts to generate time-series using GANs and VAEs, according to our survey, there is no research using SGMs for this purpose." It's not reasonable since this method may not tackle any real problems or challenges but just be proposed intentionally. It has reduced the impact of this work.

---

> ### Author Response · Authors · 2023-11-22
>
> SGMs are renowned for their generation quality and stability of learning. But there's also a problem of SGM : SGM takes lots of denoising steps, unlike other generative models which generate data one-shot, e.g., GANs. This is a fundamental issue of SGM and the main reason for the bottleneck you mentioned. And there're many prior works to solve it. We also dealt with it in the Limitation Section in Appendix of our revised paper. For your convenience, we copy the revised part below:
> > Although our method achieves state-of-the-art sampling quality and diversity, there exists a funda- mental problem that all SGMs have. That is, SGMs are slower than GANs for generating samples (see above). Since there are several accomplishments for faster sampling [1, 2], however, one can apply them to our method and it would be much faster without any loss of sampling quality and diversity.
>
> [1] Xiao. Tackling the generative learning trilemma with denoising diffusion GANs. ICLR22
>
> [2] Jolicoeur-Martineau. Gotta go fast when generating data with score-based models. CoRR21
>
> We leave applying these methods to TSGM as future work, since fast sampling is an independent topic of SGMs[1, 2].
> On the other hand, the computational cost that we mentioned was in the perspective of the score function, not of time of generating time-series. Our theorem was used to solve the high computational cost of calculating the score function and thereby train our model efficiently.
>
> Our workflow can be summarized as follow: Time-series generation is an important domain, but there were several papers using GANs and VAEs but no previos papers using SGMs. Since SGMs have surpassed other generative models in the image domain, we apply SGMs to time-series syntheses. But there is a problem: time-series must be generates considering past observations. Therefore, TSGM have to learn the conditional score function which we derive for the first time for time-series. Since directly learning the conditional score function is computationally prohibitive, we develop a more efficient way (see our main theorem) and thereby solve the computational cost problem of learning the conditional score function. As a result, TSGM can generate better data than other baselines including GANs, VAEs, etc.
>
> Lastly, we sum up the answers of two comments you mentioned:
>
> First, the high time-complexity is a rudimentary problem of SGMs. SGM’s high time complexity is due to its many denoising steps. As a matter of fact, the quality of data, the training stability, and the time complexity are a well-known trilemma of generative models. For example, GAN’s training is unstable. SGMs generate high-quality data and can be trained in a stable manner, but need longer time for generating. In Appendix H, although TSGM needs longer time than other GAN-based methods, its ratio is similar to that of image domain (cf. Table 13). We also mention that fast sampling method, which can be applied to TSGM, is an independent research subject.
>
> Second, the computation cost that we mentioned in the main paper was about calculating the conditional score function. We expected that SGMs can generate better data than GANs and VAEs, but learning the conditional score function is a novel topic in our paper. Since directly learning the conditional score function is computationally prohibitive (and this is our intention of the computation cost), we derived a denoising score matching loss of the conditional score function in Theorem 3.1 and Corollary 3.2. And by using these results to train the model, TSGM can generates better data than other baselines. Therefore, our theorems solve the computation cost problem of learning the conditional score function and thereby yield superior results.

---

> ### Comment · Reviewer_ox1H · 2023-11-23
> **The major weakness should be considered.**
>
> I think a notable work should consider most of the aspects of its effectiveness and efficiency. If the computational cost is 10K times worse than the other works, even though the performance is relatively better, maybe the score-based generative model is not appropriate for this problem. I recommend the author to address the efficiency issue, then consider publish the work. The cost is too high to apply in real-world scenarios.

---

> ### Author Response · Authors · 2023-11-23
>
> As you said, there’re notable works in the aspect of decreasing its sampling time. However, these papers pose the problem as an independent topic. In [4], a recent survey of SGMs, you can check that increasing the sampling efficiency is an independent topic of SGMs. Please refer to Section 3 of [4].
>
> Our work can be viewed as a pioneer of applying SGMs to the time-series generation domain, as [3] did. [3] recognized that the sampling time of SGMs is longer than GANs, but SGMs can generate better data and can be trained in a stable manner. And there have been follow-up works for faster sampling as independent topics. Considering this point, we think that our work can be a good/reasonable beginning point for SGM-based time series synthesis.
>
> [1] Xiao. Tackling the generative learning trilemma with denoising diffusion GANs. ICLR22
>
> [2] Jolicoeur-Martineau. Gotta go fast when generating data with score-based models. CoRR21
>
> [3] Song. Score-based Generative Modeling Though Stochastic Differential Equations. ICLR21
>
> [4] Ling. Diffusion Models: A Comprehensive Survey of Methods and Applications. ACM Computing Survey 2022.

---

### Official Review · Reviewer_KQyu · 2023-11-02

**Soundness:** 3 good
**Presentation:** 3 good
**Contribution:** 3 good
**Rating:** 8
**Confidence:** 3

**Summary:**

The paper introduces a time series generative model employing score-based models. It involves pre-trained encoder-decoder units and a score-based model operating within the encoded space. This innovative method demonstrates effectiveness with irregular time series data. The authors conducted experiments using four datasets, generating distinct scenarios to evaluate their approach against reference models.

**Strengths:**

- The authors address an intriguing problem in their research.
- The architecture and concepts appear well-aligned with the identified issues.
- The empirical assessment carried out on benchmark data demonstrates the promising potential of the proposed approach.

**Weaknesses:**

In my view, the contribution doesn't appear to be highly innovative. The authors rely on two established models, the autoencoder and the conditional score-based approach. While this combined concept demonstrates practical efficacy, it seems more evolutionary than revolutionary. Furthermore, it's worth noting that prior works also explore the use of score-based models for time series, as evidenced in the research by Tashiro, Yusuke, et al. in "Csdi: Conditional score-based diffusion models for probabilistic time series imputation," presented in Advances in Neural Information Processing Systems 34 (2021): 24804-24816. The model is designed for imputation, while the authors solve the problem of time-series generation, but the idea is still similar.

**Questions:**

I do not have any questions for authors.

---

> ### Author Response · Authors · 2023-11-16
>
> **Q1.It's worth noting that prior works also explore the use of score-based models for time series, as evidenced in the research by Tashiro, Yusuke, et al. in "Csdi: Conditional score-based diffusion models for probabilistic time series imputation," presented in Advances in Neural Information Processing Systems 34 (2021): 24804-24816.**
>
> Although there are few works, it was not for time-series generation. So we analyze them in Appendix rather than in main paper. We also treat CSDI in Appendix C.2.

---

> > ### Author Response · Authors · 2023-11-22
> >
> > Dear Reviewer KQyu,
> >
> > Could you please check our rebuttal above and let us know whether you have more concerns?
> >
> > Best, Authors.

---

### Official Review · Reviewer_Y133 · 2023-11-04

**Soundness:** 2 fair
**Presentation:** 1 poor
**Contribution:** 4 excellent
**Rating:** 5
**Confidence:** 3

**Summary:**

This work applies score-based generative methods to the time series domain. The model consists of both an RNN style encoder/decoder which transforms the time series into a latent space, and a score-based model which generates extensions to the time series. The RNN encoder/decoder allows this model to address time series with missing data in an otherwise regularly sampled data set. The authors show the generative quality of this model outperforms others through a train-synthetic-test-real and a discriminator comparison.

**Strengths:**

Using an RNN to encapsulate the temporal information both leverages the strengths of previous works and does not allow the leakage of future information. This is an expressive and robust way to create a temporal latent space which has the added benefit of handling data with missing samples.

With this encoder/decoder and a score-based generative model, this work combines two powerful architectures to generate time series that maintain data set characteristics much better than other methods. The authors demonstrated this via "discriminative" and "predictive" scores. On these metrics, the introduced model achieves state-of-the-art results with a considerable increase in the "discriminative" task.

**Weaknesses:**

## Writing
The writing of this paper is very poor and may have been submitted without being reviewed by others. After reading, I left this work with a similar amount of understanding as I would get from a short high-level conversation about a colleague's project. In general, this work is quite vague. There is a large amount of detail about this work that is missing. Below are a few

1) What does this model look like? This work is centered on a model that is not described! Instead, the authors vaguely mention that the encoder/decoder are RNNs, and the backbone is a score-based model with a VE, VP, or subVP score (without defining what these abbreviations mean when they are first mentioned). There is a tremendous lack of detail as one can imagine many ways to satisfy this statement.
* What are the $\mathbf{f}$ and $g$ used in Eq. 1? These are crucial components of the Langevin sampling that these score models are based on, but these expressions are missing! This work mentions the VE, VP, and subVP from previous works, but does not provide the expressions that this work used.

2) Much of the text in this work is superfluous. Providing all the examples is too much so I will provide a single one here

>"For synthesizing regular time series, we use a recurrent neural network-based encoder and decoder. Continuous-time methods, such as neural controlled differential equations (Kidger et al., 2020) and GRU-ODE (Brouwer et al., 2019), can be used as our encoder and decoder for synthesizing irregular time series (see Section 3.2 and Appendix I).

> When synthesizing regular time-series only, the simplest way is to synthesize matrices, where D x N means the number of features and
means the sequence length. However, our goal is to support both regular and irregular time-series, and our proposed design is more general than this approach."

* The first paragraph states that most of the explanation for this statement is given later. Still, a one or half-sentence description would be ideal so the reader gains some understanding rather than having to believe the authors.
* The second paragraph is the primary issue. It adds no new information and introduces a variable $D$ that is never used in the paper again. It is likely obvious to readers that a multivariate time series would be represented as a matrix, yet the authors decided to spend an entire paragraph describing this. Worse, the authors chose to prioritize this over expanding on vital information to understand this model (i.e. the $\mathbf{f}$ and $g$ expressions).
* The authors already say at the end of the first paragraph they are accomodating irregularly sampled time-series data, there is no need to say it again immediately after.

3) The language and layout of this work are in a way that does not consider the readers' experience. It assumes that the reader is intimately knowledgeable about both the cited works and acronyms (i.e. knows what VP means and the expression). The authors need to explain in detail the elements that are crucial to what they did.

4) The theorems and corollary are statements followed by either more statements or a description of the terms. At no point are they proven nor do the authors give some sense of why they may hold. If the authors would like to write in such a manner they should look to other works for proper examples. If the authors wish to explain this otherwise these statements still lack explanation and motivation which confuses the reader.

5) The text references many appendices, but they are all missing. Even with appendices, the authors still need to give a brief description of what is happening rather than making a statement, referencing an appendix, and moving forward without providing context.

## Score vs Diffusion models
The authors claim that this work is unique and the first of its kind, but fail to make adequate comparisons to previous works using diffusion models. Worse the authors falsely state

> "Although there exist diffusion-based time-series forecasting and imputation methods, our target score function and its denoising score matching loss definition are totally different from other baselines."

to dismiss previous works without citing any of them. In Song et. al.'s work "SCORE-BASED GENERATIVE MODELING THROUGH STOCHASTIC DIFFERENTIAL EQUATIONS" they show that diffusion models are score based models with specific expressions of $\mathbf{f}$ and $g$. Song et. al.'s work is not mentioned in this paper yet it is highly related and disproves various statements including the one above. I can imagine other interpretations of these statements, but again the authors do not follow up with detail about how their results are unique so the reader is left confused. In this light both the above and below statements need modifying since time-series generation using diffusion models does exist.

> "Despite the previous efforts to generate time-series using GANs and VAEs, according to our survey, there is no research using SGMs for this purpose. Therefore, we extend SGMs into the field of synthesis1."

**Questions:**

I find this work very interesting, but the presentation of this work is full of vague statements and lacks crucial information. This is ultimately unsuitable for publication. This paper is unable to properly portray the model, theory, and results in a way that others can adequately understand and reproduce this work. Consequently, this paper needs to be drastically changed. I believe that by making the following changes this work can be both exciting and impactful, but in its current state it is difficult to understand and consequently unconvincing.

1) Please address all of the issues and examples I mentioned in the "weaknesses" section. Most of these issues are pervasive and extend beyond the examples I gave. Without a drastic change that addresses those issues it is hard to move forward with this work.

2) Be more explicit that this method is not validated on all possible irregularly sampled data, it is only valid for data that is regularly sampled but is missing some measurements. This work only dropped points, but did not add points that are at non-integer spacings of the regular sampling period.

3) The first line under Figure 2 is important to your argument and needs more evidence and explanation to why $\mathbf{h}_n \sim \mathbf{x}_n$.

4) The appendices are missing. Without these appendices, it is very difficult to fully review this work since many of the details exist there.

5) Some typos
* In Eq. 5, $\hat{\mathbf{x}}_n$ is also an RNN so shouldn't it have the same arguments as $\mathbf{h}_n$?
* "Eq. equation 12"

---

> ### Author Response · Authors · 2023-11-16
>
> **Q1. What are the \textbf{f} and \textbf{g} used in Eq. 1?**
>
> In the Introduction section, we mentioned the type of SDEs which we used : "Our TSGM can be further categorized into two types depending on the used stochastic differential equation type: VP and subVP."
>
> We give additional section in Appendix N to explain the 3 types of SDEs. Especially, we described the detailed forms of **f** and g. They are called a drift and a diffusion respectively, which constitute a Brownian motion of adding noises into data samples.
>
> **Q2. "For synthesizing regular time series, ~ (see Section 3.2 and Appendix I)." & "When synthesizing regular time-series only, ~ and our proposed design is more general than this approach." A  one or half-sentence description would be ideal for the first paragraph. The second paragraph is the primary issue. It adds no new information and introduces a variable that is never used in the paper again.**
>
> We modify the passage you mentioned. Especially, the second paragraph seems to be unnecessary so we remove it and add some phrases describing **f** and g.
>
> **Q3. The theorems and corollary are statements followed by either more statements or a description of the terms. At no point are they proven nor do the authors give some sense of why they may hold.**
>
> We add intuitive descriptions for our loss function prior to our theorem in Section 3.3. We mention the reason of why we have to use the two loss functions and the motivation of our main theorem.
>
> **Q4. The text references many appendices, but they are all missing.**
>
> We already included our appendices in the supplementary zip file in our initial submission. Please check it and we also attach them to our revised main paper for your convenience.
>
> **Q5. The authors falsely state "Although there exist diffusion-based time-series forecasting and imputation methods, our target score function and its denoising score matching loss definition are totally different from other baselines." to dismiss previous works without citing any of them. The authors do not follow up with detail about how their results are unique so the reader is left confused. In this light both the above. "Despite the previous efforts to generate time-series using GANs and VAEs, according to our survey, there is no research using SGMs for this purpose. Therefore, we extend SGMs into the field of synthesis1."**
>
> We have justified the differences between them and ours on Appendix B. Since time series generation is different from imputation and forecasting, we did not write it in the main paper but explain it in the appendix with several experiments. We add appropriate citations in the our revised main paper. Since you missed our appendix in the supplementary zip file in our initial submission, all these confusions are caused. Sorry for not mentioning that our appendix is in the zip file in our initial submission.
>
> **Q6. Be more explicit that this method is not validated on all possible irregularly sampled data, it is only valid for data that is regularly sampled but is missing some measurements.**
>
> Time-series synthesis is utilized in data augmentation for insufficient or imbalanced training data. For example, Physionet[1], a well-known dataset for time series classification, deliberately removed 90\% of observations to preserve the privacy of patients, causing difficulties for learning and analysis.
>
> To deal with such a case, we randomly drop with various missing rates and generate total sequence by using a Neural CDE and a GRU-ode as an autoencoder pair. We specify our settings in the experiment section.
>
> [1] Ikaro Silva. Predicting in-hospital mortality of icu patients: The physionet/computing in cardiology challenge 2012. In 2012 Computing in Cardiology. IEEE, 245–248.
>
> **Q7. The first line under Figure 2 is important to your argument and needs more evidence and explanation to why $\textbf{h}_{n}  $ and $\textbf{x}_{1:n}$ are substituted**
>
> Since we use RNN and Neural CDE as autoencoder, we can substitute total sequence with a latent variable, as we mentioned in related work section, Section 2.2. In fact, the relation above is the main design philosophy (or goal) of RNNs. We add more explanations about in Section 3.3.
>
> **Q8. Some typos (1) In Eq. 5, $\hat{\textbf{x}}_n$ is also an RNN so shouldn't it have the same arguments as $\textbf{h}_n$?
> (2) "Eq. equation 12"**
>
> (1) You can intuitively think that total sequence is accumulated in latent feature and each total sequence is restored by subtracting past data from latent feature. This is derived by the autoencoder loss function, $L_{ed}$ in equation (6).
>
> The above approach is also used in TimeGAN[1].
>
> [1] Yoon, Time-series generative adversarial networks.
> Neurips19.
>
> (2) We modified it in our revised main paper.

---

> > ### Author Response · Authors · 2023-11-22
> >
> > Dear Reviewer Y133,
> >
> > We solve your comments and revise main paper.
> >
> > Could you please check our rebuttal above and let us know whether you have more concerns?
> >
> > Best, Authors.

---

> ### Comment · Reviewer_Y133 · 2023-11-22
> **Rebuttal Response**
>
> Thank you for addressing the examples of my concerns that I provided. I believe this work is now more impactful and convincing. I am consequently inclined to improve my acceptance score, but I am still sufficiently far from being convinced that this paper is ready for acceptance. While I find the work done interesting, the poor writing causes an insufficient presentation of the work that is not worthy of being published here. The authors addressed the examples I laid out, but in my review I stated that these issues of vagueness and incomplete descriptions of the claims are pervasive. The list I provided above was just some of the examples. Since the authors addressed those examples my opinion has changed, but the authors did not address my core concern of how pervasive these issues are. Consequently, my thoughts on this work have not changed enough for me to change my decision to acceptance.
>
> The authors need to read my general concerns more thoroughly and address them more completely. The examples I first provided show up in other parts of this work as well, the authors need to find them and change them. Here are two more examples: one serious and one careless:
>
> 1) In section 3.2 it is mentioned that the encoder and decoder are RNNs, but the type of RNN is not stated. Instead, the reader is referred to the appendix. This is a significant problem for the same reason I asked to provide the drift and diffusion terms. However, the authors failed to apply such changes to the remainder of the paper.\
> \
> Later, this same section states "The encoder e and the decoder d can consist of recurrent neural networks, e.g., gated recurrent units (GRUs)." So the RNN could possibly be a GRU? Or is this e.g. statement telling the reader that a GRU is a type of RNN, which would be a very obvious statement? Regardless, the reader still doesn't know what kind of RNN is being used or if multiple ones will be tested. A brief one or half-sentence explanation of what this RNN is before stating that the appendix contains the detailed architecture would provide a lot of information on this crucial component.
>
> 2) In the newly added portion under section 3.3 the "log" is italicized. Only variables should be italicized. This is a careless mistake that should have been caught by peers reviewing this paper. This example adds to my belief that a single person who is not very experienced in writing papers is the sole reviewer and writer of this work. This work can benefit from having others or more people edit it.
>
> Ultimately, the changes that were made are very positive and welcomed but they are quite insufficient to address the overall vagueness and carelessness of the writing. I am not calling for a rewrite of this paper, but the changes that I believe are needed would require at least a sentence or two of every paragraph to be altered. Below are some other concerns in regard to your previous message:
>
> Q1: The addition to the appendix will be very helpful to readers. I still believe that these terms, which are paramount to this study, belong in the main body of the paper. Any term or explanation that is crucial to the paper should be obvious to the reader. Omitting such crucial information often causes confusion. Basically, keep the appendix but also show the expressions in the body of the paper. Your discussion in the appendix should remain in the appendix.
>
> Q6: I believe this point may be more nuanced. This work does address data sets with missing data. I would like you to be more explicit about your claim of irregular sampling. If samples are taken at [1dt, 2dt, 3dt, 5dt] the model is fine. If samples are taken at [1dt, 2.37dt, 2.69dt, 4.15dt] it appears to me that the model cannot handle this. There are many problems that have irregular sampling like the latter. When you first mention irregular sampling I believe you should say it is only with respect to missing data in an otherwise regularly sampled data set.

---

> > ### Author Response · Authors · 2023-11-23
> >
> > Thank you for your kind response. We accept all your comments above but you still have some concerns. We summarize our answers to them bellow:
> >
> > 1. We specify that RNN is a type of model and GRU is an example of RNN. Also, we mention that the choice of RNN is a hyper parameter and we choose GRU as a default hyperparameter.
> > 2. We modify the typo of log likelihood.
> > 3. We move the description of exact drift and diffusion terms to our main paper.
> > 4. We also add clear descriptions that our case is with respect to missing observations in regular time-series.
> > 5. We clarify why $\textbf{h}_n \sim \textbf{x}_{1:n}$ by citing the autoregressiveness of RNNs.
> > 6. We specify how many steps TSGM needs for Forward SDE and Reverse SDE.
> > 7. We attach detailed explanations of GRU-D in Appendix O for page limit.
> >
> > We also revised several issues you worry now. Please see our revised paper.

---

### Official Review · Reviewer_thaZ · 2023-11-06

**Soundness:** 2 fair
**Presentation:** 3 good
**Contribution:** 2 fair
**Rating:** 5
**Confidence:** 4

**Summary:**

This paper proposed TSGM, a score-based generative model (SGM) for synthesizing time series data. The TSGM comprises an encoder-decoder architecture to transform data into the latent space, in which an SGM conducts the generation process. The loss function is derived from the denoising score matching (DSM) widely used in the current SGM. Notably. TSGM demonstrates superior performance when compared to competitors based on Variational Autoencoders (VAEs) and Generative Adversarial Networks (GANs).

**Strengths:**

1. The methodology and experiment are clearly written.
2. The performance of the proposed TSGM is obvious better than the previous methods in terms of time series generation.

**Weaknesses:**

1. The paper introduces TSGM for time series data synthesis. The paper highlights TSGM's superior performance in time series generation compared to VAEs and GANs. However, the paper's weaknesses include a lack of clear justification for the importance of time series generation, a lack of novelty in comparison to existing works, the absence of comprehensive comparisons for forecasting and imputation, and issues related to the clarity of theorems and corollaries.

2. The authors did not clarify why time series generation is significant. The authors mentioned in the introduce that “In many cases, however, time-series samples are incomplete and/or the number of samples is insufficient”, which is served as the motivation for this work. However, for incomplete cases, time series imputation can be used, which has been investigated in the previous works mentioned in the paper, such as TimeGrad and CSDI. For insufficient sample number case, the authors did show whether generation samples can increase the quality or robustness of learning.

3. The proposed method lacks novelty when compared to existing works like TimeGrad and CSDI. In the appendix, the authors differentiate their approach from these existing works by highlighting that TSGM is designed for synthesizing time series data from scratch, while the other works do not offer this capability. However, the authors did not discuss whether these works can be easily extended for generation time series.

4. The experiment part needs to be strengthened. Can the authors compare with these methods in terms of forecast and imputation? forecast and imputation of time series can be considered as conditional generation, which is more challenge and important.

5. The Corollary 3.2 seems useless. The operation of removing the expectation regarding to x^0_{n} is redundant, as in L2 we have taking the expectation regarding to x^0_{1:N}.

6.  The index of theorems and corollaries should be unified.

**Questions:**

see weakness

---

> ### Author Response · Authors · 2023-11-16
>
> **Q1.The authors did not clarify why time series generation is significant. The authors mentioned in the introduce that “In many cases, however, time-series samples are incomplete and/or the number of samples is insufficient”, which is served as the motivation for this work. However, for incomplete cases, time series imputation can be used, which has been investigated in the previous works mentioned in the paper, such as TimeGrad and CSDI. For insufficient sample number case, the authors did show whether generation samples can increase the quality or robustness of learning.**
>
> There are many articles on real-world applications of time series synthesis. A selected list of citations is as follow [1,2,3,4], and we put all the citations in the revised paper with appropriate discussions.
>
> Time-series synthesis serves a crucial role in data augmentation for insufficient or imbalanced training data. In addition, we can also address the privacy issues in releasing real data. For instance, Physionet[5], a famous dataset for time series classification, deliberately removed 90\% of observations to protect the privacy of patients, posing challenges for learning and analysis. We can release synthesize time series instead.
>
> The task of generating data while considering missing values, i.e., irregularity, and feature interdependences presents a considerable challenge. Our research is specifically tailored to confront this complexity.
>
> [1] S.Dash. Medical time-series data generation using generative adversarial networks. AIME, 2020.
>
> [2] M.Dogariu. Generation of Realistic Synthetic Financial Time-series. ACM, 2022.
>
> [3] Y. Kang. GRATIS: Generating Time series with diverse and controllable characteristics. ACM, 2020.
>
> [4] Q. Wen. Time series data augmentation for deep learning: A survey. ISCAI, 2021.
>
> [5] Ikaro Silva. Predicting in-hospital mortality of icu patients: The physionet/computing in cardiology challenge 2012. In 2012 Computing in Cardiology. IEEE, 245–248.
>
>
> **Q2.The proposed method lacks novelty when compared to existing works like TimeGrad and CSDI. In the appendix, the authors differentiate their approach from these existing works by highlighting that TSGM is designed for synthesizing time series data from scratch, while the other works do not offer this capability. However, the authors did not discuss whether these works can be easily extended for generation time series.**
>
> Please read Appendix B and C. Not only did we discuss the differences between previous works and TSGM in Appendix B, but we also conducted several experiments for extending TimeGrad and CSDI toward generation tasks in Appendix C. For TimeGrad, it failed to generate Energy, which is a high dimensional dataset. CSDI also synthesized poor data on Energy and AI4I.
>
>
>
> **Q3.The experiment part needs to be strengthened. Can the authors compare with these methods in terms of forecast and imputation? forecast and imputation of time series can be considered as conditional generation, which is more challenge and important.**
>
> It needs significantly modifying the architecture to conduct forecasting and imputation. Instead, we modified the methods of forecasting and imputation toward generation tasks in Appendix B and C. We demonstrate that TSGM outperforms other domains' models. Also, as we explained in Q1, time-series generation is not only a hard and significant task comparable to forecasting and imputation but also an independent task distinguished from other domains. Therefore, we leave the application of TSGM to other domains as future work.
>
> **Q4.The Corollary 3.2 seems useless. The operation of removing the expectation regarding to $\textbf{x}^0_{n}$ is redundant, as in L2 we have taking the expectation regarding to $\textbf{x}^0_{1:N}$.**
>
> Please note that the expectation of $\textbf{x}^0_{n}$ of Theorem 3.1 and the expectation of $\textbf{x}^0_{1:N}$ are independent. Therefore, we have to independently sample $\textbf{x}^0_{n}$ after sampling $\textbf{x}^0_{1:N}$. That is why $\textit{the law of total expectation}$ is used in such a case and our corollary can be viewed as an application of the law (c.f. Appendix A). Thanks to corollary, it is sufficient to sample $\textbf{x}^0_{1:N}$ only once.
>
> **Q5.The index of theorems and corollaries should be unified.**
>
> Thanks for your indication. We unify the index.

---

> > ### Author Response · Authors · 2023-11-22
> >
> > Dear Reviewer thaZ,
> >
> > We accept not only main comments, but also minor things too.
> >
> > Could you please check our rebuttal above and let us know whether you have more concerns?
> >
> > Best, Authors.

---

> > ### Comment · Reviewer_thaZ · 2023-11-23
> > **response**
> >
> > Thank you for your response.
> >
> > While the authors have incorporated papers highlighting the importance of time series generation tasks, the argument remains unconvincing. This is due to the absence of evidence demonstrating the potential improvement of downstream applications, such as regression or classification on time series data, through the generated time series. I recognize the potential significance of time series generation; however, the present content of the paper lacks the requisite significance and novelty for publication here.
> > The authors are encouraged conduct additional study on illustrating the advantages of the generated time series of the proposed algorithms for downstream applications. Alternatively, the authors should demonstrate whether the proposed methodology can be effectively employed for tasks such as imputation or forecasting.
> >
> > The necessity of Corollary 3.2 remains unclear due to the property of conditional expectation. In Eq. (22) of the Appendix, the two expectations can be directly combined without additional computation. For instance: \mathbb{E}_{p(x,y)}E_{p(x)}[f(x,y)] = \mathbb{E}_{p(x,y)} [f(x,y)].

---

> ### Author Response · Authors · 2023-11-23
>
> The downstream tasks you mentioned are different subjects. Time-sereis generation, imputation, and forecasting are independent topics. It sounds to us that you want us to build a sort of foundation model for time series, with which we can perform all possible tasks for time-series. Unfortunately, however, this had not been tried before. As a matter of fact, this had been accomplished for natural language processing only, e.g., ChatGPT. Even for computer vision, we do not have a strong foundation model that well perform all computer vision tasks. Please understand that those tasks are independent for now.
>
> For Corollary 3.2, it is similar with the law of total expectation. It seems obvious that $\mathbb{E}(X)=\mathbb{E}(\mathbb{E}(X|Y))$, but not trivial in the mathematical perspective by the point that it is called as theorem. Similarly, in the proof of Thm 3.1, you can know that the the probability of $\textbf{x}_n^0$
>
> is cumbersome to derive our target loss, $L_{score}^{\mathcal{H}}$. Therefore, although we might write its proof by an seemingly easy application of the law of total expectation, we show an expansion of the proof of Corollary 3.2.

---

### Author Response · Authors · 2023-11-16

First of all, thanks for your comments. We resubmit our revised manuscript and please check it.
The revised paper concatenate main paper and appendix to help review process.

---

### Author Response · Authors · 2023-11-21

Dear Reviewers,

This is a gentle reminder regarding our responses.

It would be a great help if you can check our response and let us know follow-up questions if any.

Best,
Authors.

---

### Meta-Review · Area_Chair_dDfG · 2023-12-13

**Metareview:**

This paper introduces a new score-based framework for time series generation. The model can handle both regular and irregular time series data, and it achieves promising results on a range of benchmark problems compared to GT-GAN, Time-GAN, and TimeVAE. While the approach is novel and relevant, several reviewers have reservations about this work. I feel that the authors have addressed most of the questions and concerns during the rebuttal phase. However, some concerns remain. First the quality of writing and clarity remains a major issue. (Even the revised parts have typos.) Second, the proposed architecture is highly over-parameterized compared to other methods for time series generation. In turn, training and inference is very slow. The paper misses to present exact details on the number of weight parameters, as well as details about training and inference time, but these details become apparent from looking at the provided research code. For instance, for the Stocks problem the proposed model is using around 15M learnable parameters, whereas other methods can get comparable results with using only around 30K to 100K learnable parameters.

In its present state, the paper shows promise but isn't ready for publication. It would greatly benefit from an additional iteration. I recommend rejecting this submission.

**Justification For Why Not Higher Score:**

This paper introduces an interesting and novel framework for time series generation. I would support accepting this paper, if the quality of writing, and clarity would be better. The amount of revisions made during the rebuttal phase indicates that this paper is not ready for publication in its current state. Moreover, details such as learnable parameters, training time and inference time should be discussed.

**Justification For Why Not Lower Score:**

N/A

---

### Decision · Program_Chairs · 2024-01-16

Reject